# Prolonged normothermic perfusion of the kidney prior to transplantation: a historically controlled, phase 1 cohort study

Richard Dumbill [1,2] ✉, Simon Knight [1,2], James Hunter[1,2,3], John Fallon [1,2], Daniel Voyce[4,5], Jacob Barrett [4], Matt Ellen[4], Elizabeth Conroy [6], Ian SD Roberts[7], Tim James [8], Gabrielle Allen[8], Jennifer Brook [8], Annemarie Weissenbacher [9], Rutger Ploeg [1,2], Constantin Coussios [4,5] ✉ & Peter Friend [1,2,4]

Kidney transplantation is the preferred treatment for end-stage renal disease and is limited by donor organ availability. Normothermic Machine Perfusion (NMP) might facilitate safe transplantation of marginal organs. NKP1 is a single centre, phase 1, 36-patient, three-stage cohort study investigating the safety and feasibility of up to 24 hours of renal NMP prior to transplantation. 30-day graft survival (primary endpoint) was 100%. Secondary objectives were assessment of the effect of NMP on post-transplant clinical outcomes and ischaemia-reperfusion injury, identification of predictive biomarkers, and characterisation of the performance of the preservation system. Clinical outcomes were comparable to a matched control cohort with 12-month estimated glomerular filtration rate (eGFR) 46.3 vs 49.5 mL/min/1.73m$^2$ (p = 0.44) despite much longer total preservation times (15.7 vs 8.9 hours controls, p < 0.0001). We saw strong correlations between biomarkers measured ex-situ and post-transplant outcomes, including graft function at one year (correlation between GST-Pi delta and 12-month eGFR, $R = 0.54$, $p = 0.001$). Renal NMP is useful for optimising logistics and as an organ assessment technique, and has potential to expand the donor pool. Trial registration number: ISRCTN13292277.

Kidney transplantation results in improved quality and quantity of life for most patients with end-stage renal failure compared to maintenance dialysis[1–3]. There is an increasing global burden of renal disease[4], a shortage of suitable organ donors[5], and the costs associated with dialysis are notoriously high[6–8]. Finding new strategies to prolong graft survival and expand the donor pool is crucial. Normothermic machine perfusion (NMP) is an emerging ex-situ preservation technique which maintains the kidney in a functioning state by circulating oxygenated perfusate through the vasculature at 37 °C outside the body[9]. Maintenance of homoeostasis is challenging due to ex-situ urine production[10], intolerance to haemolysis[11,12], the vulnerability of the cortico-medullary junction to ischaemia[13], and a propensity

[1]Nuffield Department of Surgical Sciences, University of Oxford, Oxford, UK. [2]Transplant Department, Oxford University Hospitals NHS Foundation Trust, Oxford, UK. [3]University Hospitals Coventry and Warwickshire, Coventry, UK. [4]OrganOx Ltd, Oxford, UK. [5]Institute of Biomedical Engineering, Department of Engineering Science, University of Oxford, Oxford, UK. [6]Oxford Clinical Trials Research Unit, Centre for Statistics in Medicine, Nuffield Department of Orthopaedics, Rheumatology and Musculoskeletal Sciences, University of Oxford, Oxford, UK. [7]Department of Cellular Pathology, Oxford University Hospitals NHS Foundation Trust, Oxford, UK. [8]Biochemistry Department, Oxford University Hospitals NHS Foundation Trust, Oxford, UK. [9]Department of Visceral, Transplant and Thoracic Surgery, Center of Operative Medicine, Medical University of Innsbruck, Innsbruck, Austria. ✉e-mail: richard.dumbill@nds.ox.ac.uk; constantin.coussios@eng.ox.ac.uk

towards vasoconstriction[14]. Previously-described clinical implementations have been limited to just a few hours immediately prior to transplantation and are therefore only applicable in the operating theatre[15–18].

Other groups have reported promising early-stage research into the use of NMP to assess[19] and recondition kidneys ahead of transplantation[16–18]. However, the only published randomised controlled trial comparing 1 h of end-ischaemic NMP after prior static cold storage (SCS) to SCS alone did not show any benefit[15]. This may be due to insufficient NMP duration—there is pre-clinical evidence that the greater the proportion of the cold-ischaemic preservation period is replaced with normothermic perfusion, the larger the effect[20]. It may also be the case that NMP alone is not an efficacious treatment for the reduction of pre-transplant injury. Several approaches to specific treatment of different aspects of ischaemia-reperfusion injury have previously been reported[21–24]—to realise an efficacy benefit, it might be necessary to combine NMP with targeted therapy. Finally, the process may require further optimisation—NMP is a highly multi-parametric intervention and there are substantial differences in the preservation environment generated by different implementations[25], rendering comparisons between approaches challenging.

NMP offers potential benefits in several distinct domains. Firstly, it may prove more efficacious than hypothermic preservation, defined as the ability to deliver an organ to a recipient in the same functional state as it was in whilst in the donor. Secondly, NMP may be valuable as assessment tool—there is preliminary evidence that safe transplantation of kidneys, declined on conventional clinical grounds, is possible after a period of NMP assessment[19]. Finally, NMP may be useful as a platform on which the donor organ can be treated and modified, either to prepare the graft for the in-vivo reperfusion event, which it will undergo on transplantation, or to make it more compatible with the intended recipient[26].

In this work, we show that prolonged renal NMP prior to transplantation is safe and feasible. Longer perfusion durations are necessary for renal NMP to be a clinically useful technique in any of these domains and are enabled by an automated device that makes operator-independent, unsupervised preservation possible. Safe longer perfusion times make detailed ex-situ organ assessment (e.g. through biopsy, or measurement of biomarkers), delivery of treatment, and minimisation of cold ischaemia time (CIT) possible. The median CIT varies across different healthcare systems but is typically around 16 h[27–29]. We therefore sought to develop and evaluate a normothermic machine perfusion system and protocol suitable for ex-situ preservation of deceased-donor kidneys for up to 24 h prior to transplantation.

## Results
### Trial conduct
NKP1 is completed, having met the recruitment target of 36 patients and followed all included recipients for 12 months post-transplant. Recruitment opened in December 2021 and completed in December 2022. All patients on the waiting list (n = 405 during the period) were consulted about the trial (Fig. 1). 48/50 (96%) of those patients approached consented to join the trial. One potential participant was excluded by the trial team based on clinical risk. 12 patients were recruited but withdrawn before transplantation. 10 of these withdrawals were due to standard clinical reasons precluding transplantation, unrelated to the trial. One was due to a device failure before the start of perfusion. One kidney was discarded due to markedly abnormal perfusion parameters. The flow of patients through the trial is illustrated in Fig. 1.

A circuit diagram and image of the perfusion machine, based on our previously-described prototype[10], are shown in Fig. 2. The arterial connection between the kidney and machine was made by anastomosis of the renal artery to a length of PTFE vascular graft, which was itself cannulated (Fig. 2B). Renal veins were cannulated (via an IVC

extension in all right kidneys). All kidneys were perfusable irrespective of vascular anatomy (27/36 kidneys single artery; 8/36 kidneys two renal arteries; 1/36 three renal arteries), with preservation of the Carrel patch for anastomosis.

A historical control cohort was selected by a pre-specified matching algorithm from a pool of 785 deceased-donor transplants (Supplementary Information, Fig. S1) performed in our centre in the 5 years preceding the trial. Two controls were matched to each trial participant (n = 72), on four variables—donor type, the 2019 UK Kidney Donor Risk Index (see 'Methods'), the induction immunosuppression agent used, and the CIT prior to transplantation (or normothermic perfusion). In the sections below, unless otherwise stated, all 36 cases were compared to all 72 controls (demographics and clinical outcomes); NMP characterisation data, NMP dose effect, and biomarkers are reported for the 36 cases exposed to NMP only.

### Demographics
Demographic data for the trial and control cohorts are shown in Table 1. The cohorts were well balanced, with equivalent UK Donor Risk Indices (mean ± SD: 1.36 ± 0.59 cases vs 1.35 ± 0.55 controls), proportions of DBD (death by neurological criteria) and DCD donors (44% DCD in each), and immunosuppression regimes (72% Alemtuzumab induction, 28% Basiliximab induction). CIT prior to perfusion was closely matched (median ± interquartile range (IQR): 8.57 ± 3.49 h cases vs 8.88 ± 3.16 h controls). The NMP preservation period resulted in substantially longer total preservation times in the trial cohort (median ± IQR: 15.70 ± 6.35 vs 8.88 ± 3.16 h). The median NMP time was 5.8 h (range 2.2–23.4 h). The second ischaemic period (interval between cessation of NMP and re-perfusion in the recipient, inclusive of anastomosis time), was kept short (median ± IQR: 68.41 ± 30.00 min), with three outliers where organ assessment or post-perfusion vascular reconstruction by the implanting surgeon was required, prior to starting the recipient operation. HLA mismatching (mismatch grade as defined by the UK organ allocation policy[30]) was well balanced. Whilst there was no overall significant difference for pre-transplant dialysis modality, pre-dialysis recipients were twice as prevalent in the control cohort. Where imbalances were present (donor age, donor history of hypertension, pre-dialysis status), these numerically favoured the controls.

### Primary and secondary outcomes
Primary and secondary outcomes are shown in Table 2. The primary outcome was 30-day graft survival, which was 100% in both cohorts (36/36 in the trial cohort; 71/71 in the control cohort due to one early patient death).

The spectrum of adverse events seen post-transplant was typical for the kidney transplant population (Supplementary Information, S2 Serious Adverse Events). There were no adverse events attributable to the intervention. The frequency of unrelated SAEs was numerically higher at longer perfusion durations; further analysis was precluded by low numbers (Supplementary Information, S2 Serious Adverse Events). There was one death (with a functioning graft) in the trial cohort due to an intracranial infection at 7 months. There was one graft loss at 8 months post-transplant. The graft had been retrieved from a 30-year-old DBD donor with diabetic ketoacidosis as the cause of death, and was subject to 2.4 h of NMP. Despite immediate graft function and a rapid initial fall in serum creatinine, the recipient's renal function subsequently deteriorated, and serial biopsies showed only progression of donor diabetic disease. There were two instances of renal arterial stenosis. One was a lower polar artery, which was cut at retrieval, subsequently tied, and detected on initiation of ex vivo perfusion. The kidney was removed from the device, re-flushed, and the lower polar artery anastomosed separately to the graft for perfusion. Post-operative angiography showed a stenosis of this artery, which was successfully angioplastied. The other was a technical

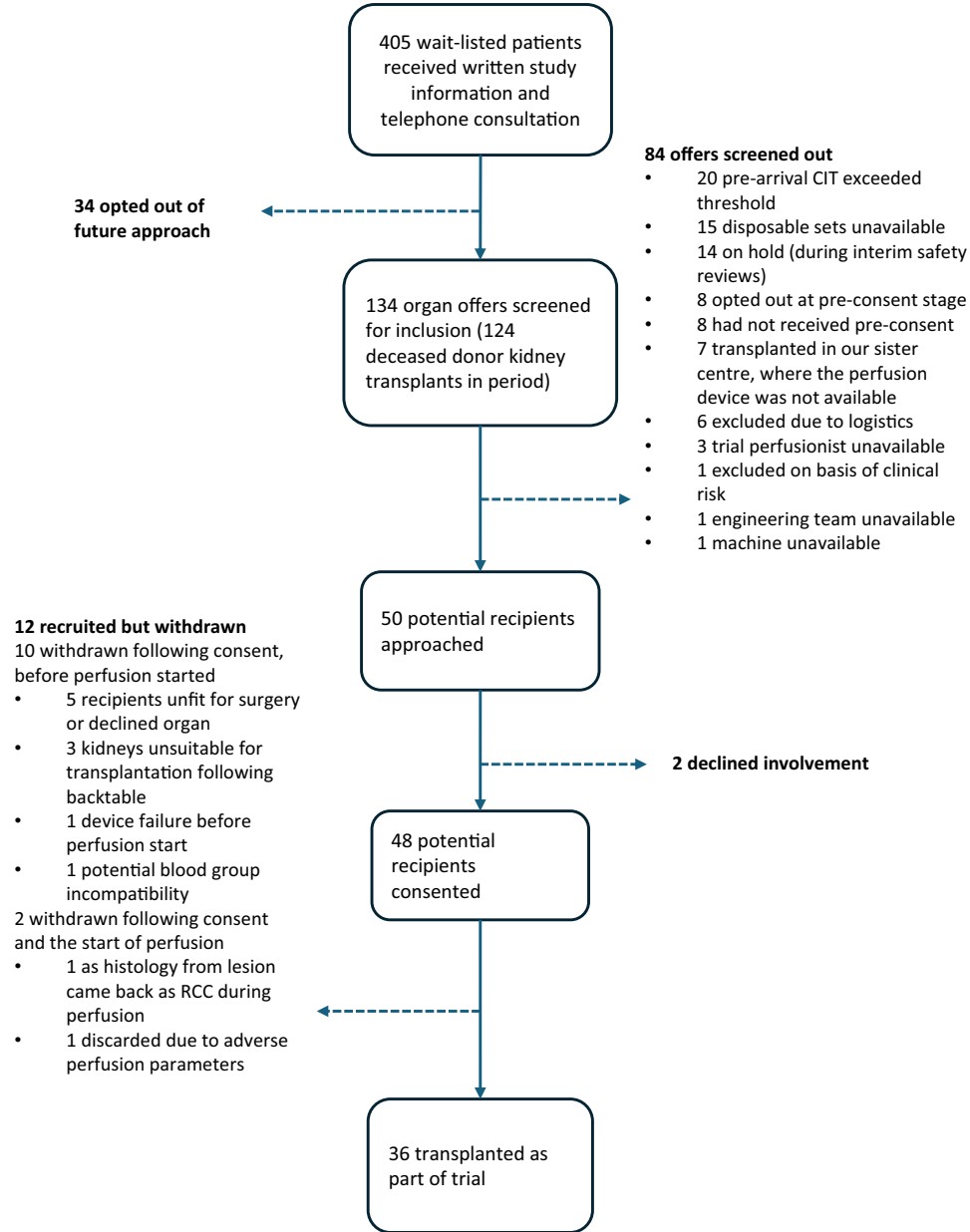

**Fig. 1 | Trial flow diagram showing the recruitment process and reasons for withdrawals or exclusions.** CIT cold ischaemic time, RCC renal cell carcinoma. The device failure before the start of perfusion was due to a software error that meant that the pump could not be started. Of the three kidneys that were unsuitable following back-benching (before the start of perfusion), one was declined has it had not been flushed adequately at retrieval, and the other two had evidence of significant arterial dissection.

complication during implantation; this was subsequently stented. There were no infectious complications related to NMP, nor evidence of microbiological contamination during perfusion (Supplementary Information, S3 NMP Microbiology).

The DGF rate (defined as any use of dialysis in the post-transplant period) was 36% in the trial cohort, vs 38% in the control cohort. This increased to 15/36 (50%) and 43/71 (61%) respectively when a functional definition of DGF, including those who were not dialysed but where serum creatinine failed to fall by at least 10% per day for the first 3 days post-transplant, was used. Graft function was also equivalent between the groups at all later timepoints (Table 2 and Supplementary Information, Fig. S4). The mean ± SD estimated glomerular filtration rate (eGFR) in the trial cohort was 41.9 ± 17.4, 46.5 ± 16.4, and 46.3 ± 19.3 mL/min/1.73 m$^2$ at 30 days, 3 months, and 12 months, respectively.

The rejection rate was noted to be low; 1/36 patients experienced a single episode of biopsy-proven acute rejection (BPAR) across the 12 months of follow-up. This occurred at 3 months post-transplant, was treated with a course of methylprednisolone, and did not recur. No patients were treated for presumed rejection without biopsy confirmation. This compares to a BPAR rate of 4/72 (5.6%) in the control cohort.

One kidney was discarded pre-transplant due to grossly abnormal perfusion parameters (Fig. 4A)– despite a normal macroscopic appearance, the renal vascular resistance was substantially higher than any other kidney we had perfused in the study. The kidney was removed, cold flushed, and NMP re-started with the same result. It was then transferred to a standard clinical hypothermic perfusion machine, which confirmed high vascular resistance, and it was discarded on these grounds. There were no device-related graft losses.

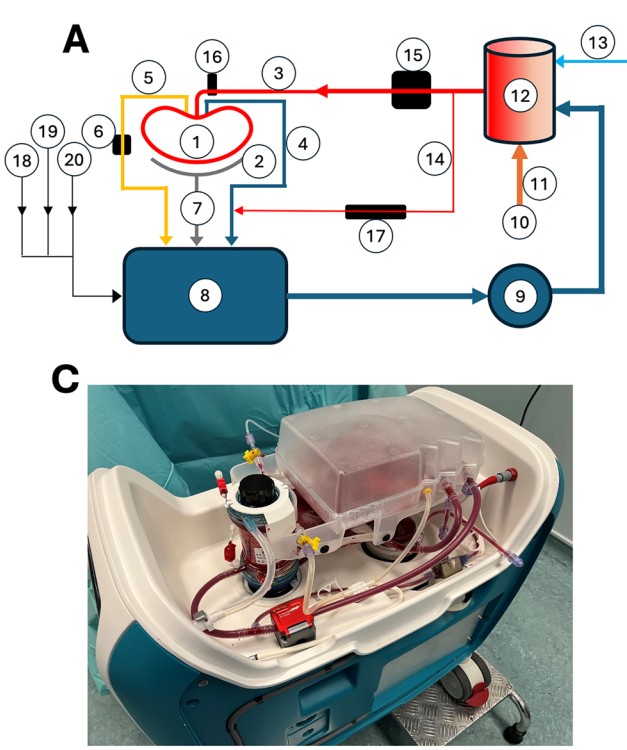
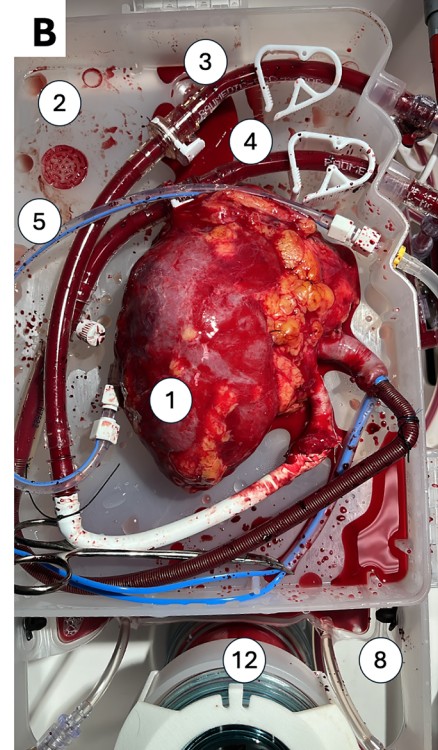

**Fig. 2 | Labelled diagram and images of the perfusion machine during use. A** schematic; **B** kidney during perfusion; **C** machine with kidney on-board. Number components−1: Kidney during perfusion. 2: Organ container. 3: Arterial blood line. 4: Venous blood line. 5: Urine line. 6: Urine flow sensor. 7: Organ container drain (passive return to reservoir). 8: Soft-shell reservoir. 9: Centrifugal blood pump. 10: Heater/cooler. 11: Heated water column. 12: Oxygenator (combined with heat exchanger). 13: Gas line in. 14: Shunt line. 15: Arterial flow sensor. 16: Arterial pressure sensor. 17: Shunt sensor. 18: Syringe driver 1 (insulin/glucose 5%). 19: Syringe driver 2 (0.45% saline initially, exchanged for glucose 20% as required). 20: Syringe driver 3 (0.45% saline initially, exchanged for epoprostenol sodium as required).

Availability of NMP during the trial was noted to alter transplant logistics, with a shift towards daytime reperfusion (Supplementary Information, Fig. S4C).

## NMP characterisation

Prior to perfusion, the perfusate was oxygenated and warmed to $36.7 \pm 1.5\,°C$. The mean connection time was $5.3 \pm 1.9$ min. At reperfusion, the temperature of the perfusate dropped to $31.4 \pm 1.3\,°C$ (Supplementary Information, Fig. S5B); it took the system $16.9 \pm 5.3$ min to rewarm the kidney to $>36.5\,°C$. This time was strongly dependent on cumulative perfusion volume (Supplementary Information, Fig. S5C). At the outset of the trial, the arterial pressure setpoint was 90 mmHg. At this pressure we frequently observed excessive renal blood flow (13/24 with steady-state renal blood flow greater than 750 mL/min), so for the final 12 cases, we decreased the target pressure to 75 mmHg. Arterial perfusate pressure was tightly controlled (Supplementary Information, Fig. S5A; perfusions 1-23 at target 90 mmHg, $87.4 \pm 5.2$ mmHg; perfusions 24−36 at target 75 mmHg, $74.8 \pm 0.2$ mmHg). There were four cases perfused at a target of 90 mmHg where this was not met due to exceptionally low vascular resistance.

Renal blood flow (RBF) was stable by 2 h into perfusion (Fig. 3A). On ex-situ reperfusion high early RBF consistently decreased to a nadir at $8.7 \pm 5.7$ min ($309 \pm 187$ mL/min), then increased to a steady-state maximum (2 h RBF $668 \pm 216$ mL/min, Supplementary Information, Fig. S5D, E). Steady-state RBF was strongly dependent on pressure (Supplementary Information, Fig. S5F). There were two notable exceptions to this pattern. Kidney no. 30 originated from a donor exposed to a cocaine overdose. It exhibited delayed vasoconstriction non-responsive to additional doses of verapamil and glyceryl trinitrate, but profoundly sensitive to epoprostenol sodium (Supplementary

Information, Fig. 5J). As described above, one kidney with a grossly abnormal renal vascular resistance was not transplanted (Fig. 4A).

Blood gases were controlled automatically throughout perfusion. The control algorithm was refined after the first 24 cases to reduce exposure to hyperoxia. Control of oxygen tension was satisfactory (Fig. 3B, first 24 cases $paO_2$ $33.3 \pm 7.1$ kPa; last 12 cases 25.5 kPa $\pm 8.8$ kPa). Carbon dioxide tension was fixed by design of the controlling algorithm (mean $4.9 \pm 0.4$ kPa); pH was allowed to vary (Fig. 3C). pH followed a consistent pattern, rising over the first 4 h before stabilising. Pre-trial experimentation had shown that commencing perfusion at a physiological pH resulted in alkalotic overshoot, which was not possible to treat without permitting $pCO_2$ to rise; therefore, our strategy was to commence perfusion at a pH of 7.1 and allow it rise to 7.35−7.45. This was successful in achieving adequate control (mean pH hour 4 onwards $7.38 \pm 0.07$).

The perfusate environment was well controlled (Fig. 3D−I). Perfusate sodium was controlled by titration of 0.45% and 0.9% saline to a target of 130-135 mmol/L. Ionised calcium consistently dropped sharply on reperfusion (Fig. 3)−a pre-perfusion target of 2.5 mmol/L resulted in a concentration during perfusion of $1.33 \pm 0.19$ mmol/L. Albumin concentration gradually fell over time (Fig. 3G). The kidneys metabolised glucose and lactate (Fig. 3H). Perfusate insulin concentration was appropriate, and gradually rose during perfusion (Fig. 3I). Oxygen consumption was static ($1.5 \pm 0.8$ mL/min/100 g, n = 35, Supplementary Information, Fig. S6H). Free haemoglobin concentration in the perfusate gradually increased over time (mean rate $0.05 \pm 0.08$ g/L/h, n = 34, Supplementary Information, Fig. S6I).

Maintenance of adequate haemodynamic, metabolic, and perfusate homoeostasis enabled successful NMP out to a maximum of 23.4 h. The distribution of preservation times is shown in Fig. 3J.

**Table 1 | Demographic data for the trial and control cohorts**

| Variable | | NKP1 (n = 36) | Control cohort (n = 72) |
|---|---|---|---|
| Timings | | | |
| First CIT, hours, median (IQR) | | 8.57 (3.49) | 8.88 (3.16) |
| Normothermic machine perfusion time, hours, median (IQR) | | 5.83 (4.54) | - |
| Second CIT, minutes, median (IQR) | | 68.41 (30.00) | - |
| Total preservation time, hours, median (IQR) | | 15.74 (6.35) | 8.88 (3.16) |
| Recipients | | | |
| Age, years, mean (sd) | | 59.2 (12.0) | 53.8 (14.4) |
| Sex, male, n (%) | | 24/36 (67%) | 46/72 (64%) |
| BMI, mean (sd) | | 28.1 (5.8) | 27.3 (5.1) |
| Dialysis modality, n (%) | HD | 22/36 (61%) | 46/72 (64%) |
| | PD | 10/36 (28%) | 9/72 (13%) |
| | Predialysis | 4/36 (11%) | 17/72 (24%) |
| Duration of dialysis, years, mean (sd) | | 2.3 (2.0) | 1.9 (2.2) |
| Previous transplant, n (%) | | 8/36 (22%) | 12/72 (17%) |
| History of diabetes, n (%) | | 12/36 (33%) | 12/72 (17%) |
| Induction immunosuppression, n (%) | Alemtuzumab | 26/36 (72%) | 52/72 (72%) |
| | Basiliximab | 10/36 (28%) | 20/36 (28%) |
| Donors | | | |
| Donor age, years, mean (sd) | | 54.0 (15.8) | 55.0 (15.1) |
| Donor type, n (%) | DBD | 20/36 (56%) | 40/72 (56%) |
| | DCD | 16/36 (44%) | 32/72 (44%) |
| Donor sex, male, n (%) | | 23/36 (64%) | 40/72 (56%) |
| Donor BMI, mean (sd) | | 28.3 (6.4) | 27.3 (5.9) |
| Donor history of hypertension, yes, n (%) | | 18/36 (50%) | 18/72 (25%) |
| Donor risk index (DRI), mean (sd) | | 1.36 (0.59) | 1.35 (0.55) |
| HLAMM group, n (%) | 1 | 0/36 (0%) | 4/69 (6%) |
| | 2 | 9/36 (25%) | 18/69 (26%) |
| | 3 | 19/36 (53%) | 28/69 (41%) |
| | 4 | 8/36 (22%) | 19/69 (28%) |

Source data are provided as a Source Data File.
*CIT* cold ischaemic time, *BMI* body mass index, *HD* haemodialysis, *PD* peritoneal dialysis, *DBD* death by neurological criteria, *DCD* donation after circulatory death, *DRI* donor risk index (2019 UK Kidney Donor Risk Index), *HLAMM* Human leucocyte antigen mismatch, *IQR* interquartile range, *SD* standard deviation.

**Table 2 | Primary and secondary outcomes, compared between the trial and control cohorts (primary indicated by bold formatting)**

| Variable | NKP1 (n = 36) | Control Cohort (n = 72) |
|---|---|---|
| 30-day graft survival, n (%) | **36/36 (100%)** | **71/71 (100%)** |
| 30-day patient survival, n (%) | 36/36 (100%) | 71/72 (99%) |
| 3-month patient survival, n (%) | 36/36 (100%) | 70/72 (97%) |
| 12-month patient survival, n (%) | 35/36 (97%) | 68/72 (94%) |
| 3-month graft survival, n (%) | 36/36 (100%) | 70/70 (100%) |
| 12-month graft survival, n (%) | 34/35 (97%) | 66/68 (97%) |
| DGF (dialysis, any) - incidence, n (%) | 13/36 (36%) | 27/72 (38%) |
| fDGF - incidence, n (%) | 18/36 (50%) | 43/71 (61%) |
| Day 2 CRR, mean (SD) | 0.16 (0.20) | 0.15 (0.24) |
| PNF, n (%) | 0/36 (0%) | 0/70 (0%) |
| 30-day eGFR, mean (SD) | 41.9 (17.4) | 43.2 (21.8) |
| 3-month eGFR, mean (SD) | 46.5 (16.4) | 49.1 (18.8) |
| 12-month eGFR, mean (SD) | 46.3 (19.3) | 49.5 (18.6) |
| Creatinine gradient (month 3–12), mean (SD) | −2 (46) | −3 (34) |
| 30-day proteinuria, n (%) | 22/31 (71%) | 18/30 (60%) |
| 3-month proteinuria, n (%) | 16/29 (55%) | 16/35 (46%) |
| 12-month proteinuria, n (%) | 6/17 (35%) | 4/11 (36%) |
| Biopsy-proven acute rejection within 12 months - incidence, n (%) | 1/36 (2.8%) | 4/72 (5.6%) |

*NS* not significant, *DGF* delayed graft function, *HD* haemodialysis, *fDGF* functional delayed graft function (any use of dialysis in the first 7 days post-transplant or failure of creatinine to fall >10% per day for the first three days), *CRR* creatinine reduction ratio, *PNF* primary non-function (note the denominator for the control cohort is 70 due to two patient deaths within 90 days whilst still dialysis dependent), *eGFR* estimated glomerular filtration rate. Where relevant outcomes are censored for patient survival (i.e. denominators are based on those alive at the follow up time). Data was complete for the trial cohort. For the control cohort, survival data was complete; eGFR was missing for two patients at 30 days (2.8%), four patients at 3 months (5.6%), and 17 patients at 12 months (23.7%). DGF data was complete; fDGF data was missing for one patient (1.4%), and CCR2 data was missing for 18 patients (25%). PNF data was missing for two patients (2.8%). For categorical data, data completeness is indicated in the table. Source data are provided as a Source Data File.

## NMP dose effect

Exploratory analyses were performed examining the effect of NMP duration on outcome (Supplementary Information, Fig. S6). Kidneys with improvement in histological appearances of the tubules had longer average perfusion durations than those where the tubules deteriorated (8.9 h vs 4.8 h, n = 18, p = 0.048; Supplementary Information, Fig. S6A). Preservation times were not significantly different between DGF groups (median 8.7 h, DGF vs 5.0 h, primary function, p = 0.2; NMP proportion: 41.5% vs 37.6%, p = 0.34; total preservation time: 18.6 h vs 15.0 h, p = 0.054, Fig. S6B–D). There was a significant correlation between post-transplant day two serum NGAL and both perfusion duration (p = 0.04) and total preservation time (p = 0.007) (n = 32, Fig. S6E–G). There was no relationship between later graft function and perfusion duration, NMP proportion, or total preservation time (Fig. S6H–J).

An additional exploratory analysis was conducted against a secondary control cohort, selected from the pool of controls using the same algorithm as for the primary control cohort but with matched total preservation time, rather than pre-perfusion CIT. Demographic data is shown in Supplementary Information, Table S7. As for the primary control cohort, baseline variables were well matched. The mean CIT (and total preservation time) in this secondary control cohort was 16.8 ± 4.9 h. Again, there were no significant differences across all examined secondary endpoints between the cohorts (Supplementary Information, Table S8 and Fig. S4).

## Biomarkers

One kidney perfused as part of the trial was discarded based on an abnormally elevated vascular resistance. There was a numerical early difference in vascular resistance between those kidneys which resulted in functional delayed graft function (fDGF) and those which functioned immediately (68.3 ± 43.0 versus 51.4 ± 30.4 mmHg 100 g mL$^{-1}$ min$^{-1}$, p = 0.08, Fig. 4A and Supplementary Information, Fig. 9A, B). Figure 4B, C shows a clear and significant difference in weight-adjusted vascular resistance between kidneys from donors with a history of hypertension and those without, and those with histopathological findings of donor vascular disease (arterial fibroelastosis) compared to those without.

The urine production rate at 2 h was numerically but not significantly better in the immediate function group compared to those with fDGF (20.4 ± 72.1 mL/h vs 15.3 ± 47.3 mL/h, p = 0.14, Fig. 4D), as was fractional excretion of sodium (4.5 ± 6.1% vs 9.0 ± 22.1%, n = 32, p = 0.07, Supplementary Information, Fig. S9F) and creatinine clearance (2.7 ± 6.9 mL/min vs 1.2 ± 2.5 mL/min, p = 0.048, Supplementary

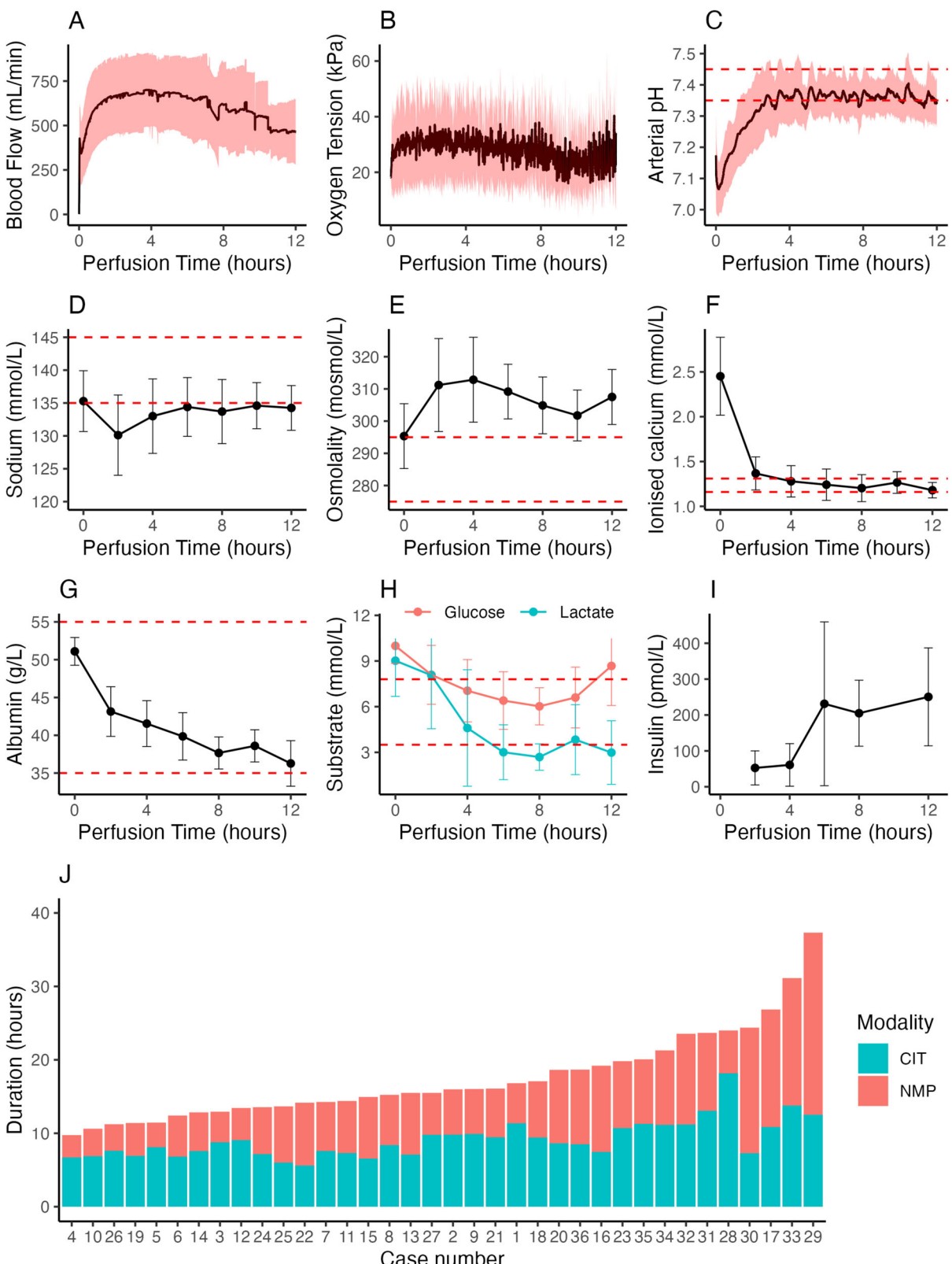

**Nature Communications** | (2025)16:4584

Information, Fig. S9D). Creatinine clearance was not associated with later graft function (Supplementary Information, Fig. S9E). We noted that 11/12 cases with an ex-situ creatinine clearance of greater than 3 mL/min resulted in either immediate function or only a single session of post-transplant dialysis. Ex-situ proteinuria appeared to be associated with better graft function at 3 months (deindexed eGFR 60.8 ± 15.4 mL/min in the high-proteinuria group versus 45.4 ± 15.7 mL/

min in the low-proteinuria group, n = 31, p = 0.01, Supplementary Information, Fig. S9H), but not at 12 months. Oxygen consumption did not discriminate between those that developed DGF and those that functioned immediately (Supplementary Information, Fig. S9I).

Renal injury markers measured in perfusate were strongly associated with post-transplant outcomes. The strongest correlations were seen with glutathione serum transferase (GST-Pi, hour 2 concentration,

**Fig. 3 | Perfusion data characterising the intervention.** For subplots (**A**–**C**), the black line shows the mean value for all cases perfused to that time point; the shaded area shows standard deviation. Horizontal red dashed lines indicate normal physiological ranges. For subplots (**D**–**I**), points indicate means, error bars standard deviations. **A** Renal blood flow during perfusion. **B** Oxygen tension during perfusion. **C** Arterial pH during perfusion (normal range 7.35–7.45). **D** Sodium concentration during perfusion (normal range 135–145 mmol/L). **E** Osmolality during perfusion (normal range 275–295 mosmol/L). **F** Ionised calcium during perfusion (normal range 1.16–1.31 mmol/L). **G** Perfusate albumin concentration during perfusion (normal range 35–55 g/L). **H** Perfusate glucose and lactate concentrations during perfusion (glucose normal range 3.5–7.8 mmol/L). **I** Perfusate insulin concentration during perfusion. For subplots (**A**–**I**) data are shown up to 12 h only, due to the small number of kidneys perfused for greater than 12 h (number of kidneys per time-point: 0 and 2 h, n = 36; 4 h, n = 27; 6 h, n = 16; 8 h, n = 11; 10 h, n = 6; 12 h, n = 4). **J** A summary chart showing the total preservation durations, divided into cold ischaemic (CIT) and normothermic machine perfusion (NMP) times. The maximum NMP duration was 23.4 h, resulting in an overall preservation time of 37.3 h. Source data are provided as a Source Data File.

immediate function group 46 ± 25 ng/mL versus 109 ± 40 ng/mL in the fDGF group, n = 35, $p = 6.1 \times 10^{-6}$, Fig. 4G, remained $p < 0.001$ after adjustment for multiple comparisons), liver-type fatty acid binding protein (L-FABP, median hour 2 concentration, immediate function group 106 ± 100 ng/mL versus 369 ± 641 ng/mL in the fDGF group, n = 35, $p = 0.002$, Fig. 4H), and interleukin 18 (IL-18, mean hour 2 concentration, immediate function group 31 ± 16 ng/mL versus 49 ± 17 ng/mL in the fDGF group, n = 35, $p = 0.002$, Fig. 4I). Lactate dehydrogenase (LDH) was also significant on unadjusted analysis (mean hour 2 concentration, immediate function group 1095 ± 529 IU/L versus 1628 ± 654 IU/L in the fDGF group, n = 35, $p = 0.01$, Fig. 4F). Interestingly, neutrophil gelatinase-associated lipocalin (NGAL) was poorly associated with fDGF (mean hour 2 concentration, immediate function group 9.2 ± 5.0 ng/mL versus 12.1 ± 6.2 ng/mL in the fDGF group, n = 35, $p = 0.14$, Fig. 4E). The between-group difference had slightly increased by the end of perfusion (median 8.0 ± 6.1 vs 11.6 ± 12.4 ng/mL, $p = 0.03$, Supplementary Information, Fig. S10E), however its discriminative ability remained relatively poor. Elevated kidney injury molecule 1 (KIM-1) appeared to be associated with better post-transplant function (Supplementary Information, Fig. S10F, G).

GST-Pi delta was significantly correlated with graft function (deindexed eGFR) at all timepoints (30 days: $R = 0.47$, n = 35, $p = 0.005$; 3 months: $R = 0.46$, n = 35, $p = 0.005$; 12 months: $R = 0.54$, n = 33, $p = 0.001$, Fig. 4J, remained $p = 0.047$ after adjustment for multiple comparisons). A linear model containing delta GST-Pi and donor age as predictors, and deindexed 12-month eGFR as the response variable suggested that delta GST-Pi and donor age were independent and of equal importance in determining 12-month graft function (n = 33, $p$-value for an interaction term 0.21; univariate $R^2$ 0.29 for GST-Pi delta; univariate $R^2$ 0.29 for donor age; multiple $R^2$ 0.48 for the full model with $p = 0.003$ for both coefficients). Additional analyses exploring GST-Pi delta as a discriminant for distinguishing good graft function from poor were performed by constructing ROC curves and determining threshold values for GST-Pi, defining good graft function as a deindexed eGFR of greater than 45 mL/min. There was a highly significant difference in delta GST-Pi between these groups at all timepoints (30 days: mean for good function group 1.8 vs −37.4 poor function, n = 35, $p = 0.0005$; 3 months: −1.2 vs −38.4, n = 35, $p = 0.0023$; 12 months: −4.5 vs −35.0, n = 33, $p = 0.009$). The GST-Pi delta cut-off values for distinguishing good function from poor were −11.8 ng/mL (AUC 0.85) at 30 days, −32.8 ng/mL (AUC 0.83) at 3 months, and −6.9 ng/mL (AUC 0.79) at 12 months (Supplementary Information, Fig. S10H, I).

## Discussion

We successfully transplanted 36 kidneys after between two and 24 h normothermic machine perfusion following initial SCS, with a 100% 30-day graft survival rate. Our greatest pre-transplant NMP time was 23.4 h, with a total preservation time of 37.3 h. This represents the longest ex vivo NMP preservation prior to clinical kidney transplantation reported in the literature to-date. We were able to achieve this safe duration of preservation by carefully controlling the perfusion environment, and as far as possible, ensuring that all measured parameters were kept within physiological ranges. Our primary control cohort consisted of a group of closely matched transplants performed at our centre in the 5 years preceding the trial, where cold ischaemia time (either pre-perfusion in the trial cases, or the total preservation duration in the control cohort) was used as a matching variable. We observed no significant differences between our trial cohort and these controls across any of our measured secondary endpoints, and there were no device-related adverse events. We chose to match our primary control cohort using cold ischaemia time rather than total preservation time, as the purpose of our trial was to evaluate safety and feasibility, rather than to test efficacy, as this would require a larger sample size than is appropriate for a phase 1 trial. Our data suggest that prolonged normothermic machine perfusion of the kidney is indeed both safe and feasible.

NKP1 was neither designed nor powered to examine efficacy endpoints, and unsurprisingly, we did not see any improvement in graft function with NMP. This result is concordant with that obtained by Nicholson et al. in their randomised trial of 1 h of end-ischaemic NMP[15]. We also found no evidence that longer perfusion times (or greater NMP proportion) were associated with better or worse clinical outcomes. There was no statistically significant association between perfusion duration and DGF, or with longer-term graft function, but there was a positive correlation between post-transplant day 2 NGAL and perfusion duration. In an exploratory analysis comparing our trial cohort to a group matched on total preservation time there were again no significant differences in clinical outcome. This result is more surprising, as in this comparison, cold ischaemia is replaced with NMP. However, it should be interpreted with caution as whilst there were no observed differences in baseline parameters between these cohorts, this control cohort is vulnerable to unmeasured selection bias—in general, clinicians seek to minimise cold ischaemia time, particularly for higher-risk kidneys. In summary, whilst we found no evidence of an adverse effect of prolonged NMP, we also found no evidence that NMP alone improves clinical outcomes.

This result does not align with the preclinical results obtained by Selzner et al., who have extensively investigated replacement of cold ischaemia with normothermic perfusion using a porcine autotransplant model[20,31,32]. The degree and type of injury are likely to be different with kidneys taken from healthy juvenile animals compared to expanded-criteria human organ donors. Secondly, none of the animals in the Selzner experiments experienced clinical DGF in that they were not dialysed post-transplant. It is therefore possible that NMP is less injurious than hypothermic preservation—sufficient to produce a difference in a highly controlled study using healthy donor animals—but the difference is of insufficient magnitude to manifest a clinical difference in a small, historically-controlled phase 1 trial in humans. Another possible reason for our failure to observe a difference in clinical outcome may be related to the initial SCS preservation phase—the mean cold ischaemia time before perfusion was started was over 9 h. Taking the device to the donor site and transporting the kidney during NMP to eliminate this initial period of cold ischaemia should be evaluated in the future.

Our biomarker data is exciting and opens the door to a raft of new ex-situ diagnostics and interventions. Of particular significance is our finding that the change in GST-Pi concentrations in the perfusate during perfusion was strongly linked with clinical outcome, even out to a year post-transplant, as graft function at 1 year is known to be of

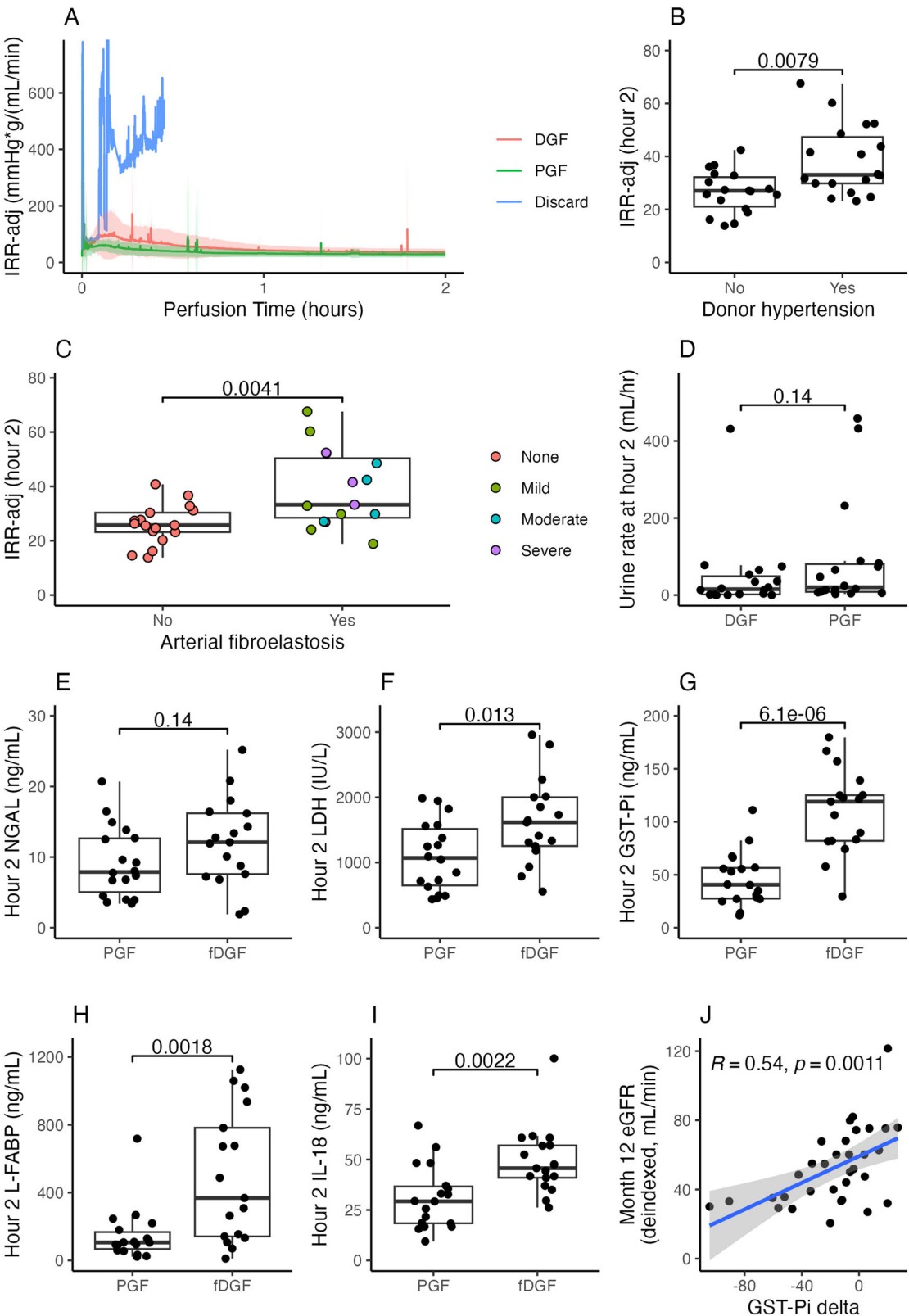

substantial prognostic value[33]. GST-Pi has previously been reported to be associated with DGF in hypothermic perfused kidneys[34], although only with moderate predictive capacity and with no relation to longer-term outcomes. In contrast, we saw a very strong association between DGF and GST-Pi early in perfusion, and a robust association between the change in its concentration and 12-month function. It is important to note that the directionality of this relationship with early and late outcomes was opposite—high levels of GST-Pi early in perfusion were related to a higher probability of DGF, however, sustained concentrations of GST-Pi in the perfusate during perfusion were related to better function at 12 months. This observation is likely to be due to GST-Pi's role as an antioxidant molecule. In previous hypothermic perfusion studies, elevated perfusate GST-Pi correlated with DGF[35,36]. In a recent proteomic analysis, the NRF2-mediated oxidative stress

**Fig. 4 | Biomarkers. A** Vascular resistance (adjusted for the weight of the kidney) during the first 2 h of perfusion, stratified by early functional status (DGF Delayed Graft Function, PGF Primary Graft Function), and showing the profile of the single kidney which was discarded based on perfusion parameters. Solid lines indicate group means; shaded areas indicate mean ± SD. **B** Weight-adjusted vascular resistance compared between kidneys from donors with a history of hypertension, and without (Wilcoxon signed-rank test, two-sided, n = 36). **C** Weight-adjusted vascular resistance compared between kidneys with histopathological evidence of donor vascular disease, and without (Wilcoxon signed-rank test, two-sided, n = 36). **D** Ex-situ renal function (urine output shown; see Supplementary Information for creatinine clearance and fractional excretion of sodium) was not significantly different between the functional DGF (fDGF) and PGF groups (Wilcoxon signed-rank test, two-sided, n = 36). **E** Hour 2 perfusate NGAL concentration was not significantly different between the fDGF and PGF groups, however perfusate LDH, GST-Pi, L-FABP, and IL-18 (subplots **F**, **G**, **H**, **I**) all showed significant differences (T-tests other than L-FABP, Wilcoxon signed-rank test, all n = 35, all two-sided). For GST-Pi in particular the difference was large and highly statistically significant, even after adjustment for multiple comparisons. Boxplots B-I: centre lines shown are medians; bounds of the boxes are first and third quartiles; whiskers extend to either the most extreme value or the limit of the box ± 1.5x IQR, whichever is closest to the median; individual datapoints are overlaid and jittered. **J** Relationship between the change in GST-Pi concentration observed in perfusate during perfusion, and post-transplant renal function (linear regression, Pearson's correlation coefficient shown, n = 35, shaded area indicates 95% CI). This association was preserved out to 12 months and remained significant after adjustment for multiple comparisons. Source data are provided as a Source Data File.

response (of which GST-Pi is a component) was substantially impaired in kidneys which went on to develop long-duration DGF[37]. These observations are all consistent with our results. Our overarching hypothesis is that GST-Pi concentrations at the start of normothermic perfusion or during hypothermia (i.e. in the absence of ongoing translational or secretory activity) are indicative of the burden of oxidative stress and injury experienced by the kidney up to that point, whereas the evolution of GST-Pi during normothermic perfusion provides information about the organ's ability to respond to a controlled, standardised reperfusion event. The ability to sustain this response during perfusion appears to be associated with improved longer-term clinical outcome. The NRF2-mediated response to oxidative stress is therefore a promising target for ex-situ modulation.

Our observation of a fall in perfusate lactate during perfusion is interesting. Other investigators have variously reported lactate to fall[38], stay constant[18], or rise[10,16,39] during perfusion. There are several possible explanations for this. Firstly, it could be the case that the cortex is under-perfused in some implementations of renal NMP. An intra-renal Cori cycle has been proposed where lactate is produced through glycolysis in the relatively hypoxic medulla and transported in the vasa recta to the cortex, where it is used in gluconeogenesis[40]. We noted that overall renal blood flow in our trial was substantially higher than that reported by other investigators[10,15,16]. It is possible that lactate rises if the renal cortex is hypo-perfused. Secondly, the balance of supplied metabolic substrate and insulin concentration might be important in determining net lactate production. We did not supply amino acids, which differs from other described protocols. Further work needs to be done to elucidate whether this difference in lactate behaviour is due to differences in metabolic substrate or regulation of metabolism. Finally, previous studies are heterogenous with respect to the cold ischaemia time and the degree of renal injury before the start of NMP, with one report that in a porcine model, the rate of lactate clearance correlated with post-transplant renal function[38]. It may be the case that our cohort of donors and relatively short pre-perfusion cold ischaemic times underlie our observed fall in lactate.

A limitation of this study is the comparison to SCS, rather than to hypothermic machine perfusion (HMP). HMP uptake in our centre, and across the UK, has been low—primarily for logistical reasons. Several studies have previously demonstrated the superiority of continuous HMP to SCS[41–43], however interestingly, oxygenated HMP after initial SCS was not shown to yield benefit compared to SCS alone – prolonged HMP appears to be necessary[44]. It may be the case that the same is true with NMP. Our results are generalisable to a broad population of donors and recipients, however, are limited by being a single-centre study with one principal perfusionist. Anastomosis between the renal arteries and a length of vascular graft for perfusion maximises preservation of the donated arterial vasculature but is technically complex and time consuming—in future, this could represent one option in a toolbox of techniques available to achieve arterial connection. Reproducible implementation of a complex technique is challenging and may be best achieved with hypothermic retrieval and subsequent NMP in specialised centres, with pre-perfusion times kept to a minimum.

In conclusion, prolonged normothermic perfusion of the kidney is safe and feasible with adequate maintenance of ex-situ homoeostasis. In the context of a phase 1 trial not powered for efficacy, NMP after initial SCS did not measurably reduce DGF. However, NMP may be useful as an organ assessment technique and as a platform for delivering ex-situ therapy. Replacement of a greater proportion of the total preservation time with normothermia by taking the machine to the donor should be evaluated, as should the delivery of targeted ex-situ therapy pre-transplant. Further work needs to be done to establish how best to assess the viability of donated kidneys ex-situ.

## Methods
### Trial design
NKP1 was a single centre, phase 1, non-randomised, historically controlled trial evaluating the safety and feasibility of prolonged duration normothermic perfusion of the kidney prior to transplantation, and complies with all relevant ethical regulations. The trial was funded by the National Institute for Health Research and sponsored by the University of Oxford. OrganOx Ltd was a named collaborator and was responsible for the design and delivery of the perfusion machines. The trial protocol was prospectively registered online (ISRCTN13292277) and is publicly available here: https://doi.org/10.1186/ISRCTN13292277. Ethical approval was granted by the Greater Manchester South research ethics committee (20/NW/0442), and the trial was overseen by the Medicines and Healthcare Products Regulatory Authority as a phase 1 device trial (CI/2021/0034/GB). The study was hosted by Oxford University Hospitals NHS Foundation Trust, overseen by the University of Oxford Clinical Trials Research Unit (OCTRU—a UKCRC-registered clinical trials unit), and supported by the Oxford University Surgical and Interventional Trials Unit (SITU). The first patient was recruited on 08 Dec 2021 (withdrawn before the study intervention was applied, see Fig. 1). The first recruited and treated patient was recruited on 01 Jan 2022, and the last patient on 12 Dec 2022.

The trial was divided into three stages, each of 12 perfusions after prior static cold storage (SCS), with incrementally increasing maximum permissible perfusion times and interim safety reviews by an independent data and safety monitoring committee between each stage. The Trial Steering Committee (composed of 5 independent members, including a patient representative, plus the Chief Investigator) authorised progression to the next stage at each interim review. The minimum NMP time at each stage was 2 h, which reflects the time required to anaesthetise the recipient and perform the first part of the operation. The maximum perfusion time for stage 1 was 6 h, for stage 2 was 12 h, and for stage 3 was 24 h. The actual perfusion duration in each case was logistically determined and performed at the discretion of the transplanting surgeon. Transplants were not artificially delayed solely for the purpose of perfusion, although overnight operating was generally deferred as our experience and maximum permitted perfusion

time increased. Very long duration perfusions occurred typically in situations where there were multiple concurrent organ offers, and/or last-minute changes of recipient. There was one substantial protocol amendment during the trial, implemented between stages 2 and 3 (REC authorisation 04 Jul 2022; MHRA authorisation 24 Aug 2022). This included a change to the permissible pre-perfusion cold ischaemia time (increased from 10 h for all kidneys to 12 h for kidneys from DCD donors, and 18 h for kidneys from DBD donors); a change in the anti-fungal used during perfusion from micafungin to caspofungin; and a decrease in the NMP perfusion pressure from 90 mmHg to 75 mmHg.

All potential recipients on the waiting list at the Oxford Transplant Centre were consulted about the trial in advance, to ensure that they had at least 24 h to contemplate the study information before deciding whether to join. This dissemination of information step included provision of a written information sheet and a telephone consultation; it was a requirement that this step had been completed for potential participants to be approached regarding the study. Appropriately consented adult (aged 18 and over) recipients were eligible to join the study. Donor inclusion criteria were kidneys from deceased donors aged above 16 years; DCD or DBD donors; accepted for transplantation according to local criteria; and cold ischaemia time (CIT) prior to NMP no greater than 12 h for DCD donors, or 18 h for DBD donors (initially this was 10 h in all cases, however due to the restrictive effect this had on recruitment the CIT limit was subsequently increased to 12 h for DCD kidneys and 18 h for DBD kidneys, as described above). Recipient inclusion criteria were male or female, aged 18 years or older; on the waiting list for kidney transplantation at the Oxford Transplant Centre, Oxford; had provided informed consent for participation in the study; able and willing to comply with all study requirements (in opinion of investigator or deputy); and fit to proceed with kidney transplantation. Donor exclusion criteria were kidneys that would not be accepted according to local criteria; kidneys accepted as a pair for dual transplant; and CIT greater than 12 h prior to initiation of NMP for DCD donors, or 18 h for DBD donors (initially 10 h, as detailed above). Recipient exclusion criteria were not willing or unable to provide informed consent; recipients aged less than 18 years; participation in an investigational study likely to affect interpretation of the trial data; undergoing living donor kidney transplantation; undergoing dual kidney transplantation; undergoing transplantation of other organ(s) in addition to the kidney.

## Control cohort

The control cohort consisted of matched historical controls (two controls per trial participant) who underwent deceased donor kidney transplantation between 01 Oct 2016 and the start of the trial recruitment period, at the Oxford Transplant Centre. The matching algorithm was specified in advance (Appendix A4 in the registered protocol), and included donor type, donor risk index (calculated as described below), the induction immunosuppression agent, and CIT (in the trial group, this was the interval from cold-flush in donor to ex-situ reperfusion on the perfusion device; in the control cohort, this was the interval from cold-flush in the donor to reperfusion in the recipient). Cases were matched to controls strictly on donor type (DBD/DCD) and induction immunosuppression (Basiliximab/Alemtuzumab), and then by minimising the summed percentage difference ('match score') in cold ischaemia time (CIT) and donor risk index (DRI):

$$Match\ score = \frac{CIT_{case} - CIT_{control}}{CIT_{case}} + \frac{DRI_{case} - DRI_{control}}{DRI_{case}} \quad (1)$$

Match scores were computed for all possible case-control pairs. The final case-control allocation was then determined as the combination which minimises the total match score summed across all cases.

This was computed in R (Version 2023.06.0 + 421) as detailed in the protocol.

An additional, exploratory analysis was conducted using a control cohort matched in the same manner as described above other than the preservation time used for matching was the total preservation time (i.e. the interval from cold-flush in the donor to reperfusion in the recipient in both groups).

De-identified data concerning the control cohorts were extracted from a local clinical database. Where data was missing as the patient had moved away from our centre, data concerning patient and graft survival, and graft function, were extracted from the national registry. If data was missing from the local clinical database and national registry, it was treated as truly missing; no imputation was performed.

## Perfusion device and protocol

A novel perfusion device was designed and constructed by OrganOx Ltd, based on a previous prototype that had been shown to enable normothermic preservation of discarded human kidneys with urine recirculation for up to 24 h[10]. This is illustrated in Fig. 2. In brief, perfusate is pumped by a centrifugal blood pump from a reservoir, through a hollow-fibre oxygenator which warms it, adds oxygen, and removes carbon dioxide. The system is pressure-controlled and includes a continuous inline blood gas sensor that permits automation of the delivery of oxygen and air to the oxygenator, to control the partial pressures of oxygen and carbon dioxide in the perfusate. The target arterial pressure was 90 mmHg at the start of the trial, later decreased to 75 mmHg (see 'Results'). The device automatically logs second-by-second data for each case (including measured pressure, temperature, renal blood flow, arterial oxygen and carbon dioxide tensions, pH, and urine flow).

The machine was primed with perfusate whilst conventional backtable surgery was performed. 100 mL 20% human albumin was added to the perfusion machine, followed by 100 mL 0.45% sodium chloride, and 1 unit of packed red cells. This was supplemented with 125 mg meropenem, 25 mg micafungin (later substituted for 12.5 mg caspofungin due to a supply shortage), 2.5 mg fondaparinux, 2.5 mg verapamil, and 1.8-2 mL 10% calcium chloride. A blood gas was taken and analysed (ABL90-FLEX, Radiometer), and a calculator used to determine the ratio of 0.45% to 0.9% saline required to achieve a starting sodium concentration of 130 mmol/L with an additional 50 mL of crystalloid. Pre-trial experimentation had revealed a consistently high osmolar gap, and so this approach balanced mild hyponatremia (133 ± 5.5 mmol/L, Fig. 3D) against mild hyperosmolality (306 ± 13.5 mosmol/Kg, Fig. 3E). Sodium bicarbonate was added as required during priming to achieve a pH of 7.1 at 37 °C. A second 'priming' blood gas was then taken, and a calculator again used to determine the ratio of 0.45% to 0.9% saline to use to prime the arterial and venous cannulas during organ connection.

The perfusion device includes three syringe drivers, all pre-programmed to run at 1 mL/h. A combined 5% glucose and insulin infusion (2–3 units of insulin diluted in 30 mL 5% glucose running at 1 mL/h) was provided from the start of perfusion, alongside two additional 30 mL syringes of 0.45% saline. When glucose was observed to be falling (usually at 4 h), one of the 0.45% saline infusions was substituted for 20% glucose. The third infusion channel remained as 0.45% saline for the duration of perfusion unless problematic vasoconstriction was encountered, in which case it was substituted for epoprestenol sodium. Glyceryl trinitrate was available as an alternative vasodilator if required.

Once conventional bench surgery had been completed, additional steps were performed to prepare the kidney for perfusion. Kidneys were weighed. A stapled venous extension was performed in all right-sided kidneys. A 5–0 prolene suture was placed at each corner of the vein and tied with a 1-cm loop, through which a 1 silk tie was passed and tied down onto a 20–22Fr wire-wound right-angled venous cannula

(Medtronic Ltd) placed into the renal vein. This kept the tip of the cannula at the correct depth without traumatising the endothelium. The venous connection was made watertight with a silicone vessel snugger. The arterial connection was made in all cases by anastomosis of all renal arteries (wherever possible via a wide aortic patch) to a length of vascular graft (7 or 8 mm diameter Venaflow or Carboflow, Bard Ltd; 6–0 Goretex suture) which was itself cannulated with a ¼" barb connector.

The kidney was then taken out of ice and placed in the sterile, draped, organ container. The cannulas were primed, and the connection made. Once re-perfused, the ureter was cannulated and connected. Urine was recirculated for all perfusions, as previously described[10].

### Transplantation

Unless there was a specific indication for back-table examination post-perfusion, before the intended recipient was anaesthetised, the kidney was taken off the perfusion machine and cold-flushed once the first stages of the implant operation had been completed, and the recipient vessels were being prepared for anastomosis. Transplantation followed usual clinical practice, at the discretion of the implanting surgeon. Standard immunosuppression was provided in line with our unit's policies.

### Sampling

Biopsies were taken pre-perfusion (24/36 cases) and intra-operatively post-reperfusion in the recipient (26/36 cases). The decision to take a biopsy was at the discretion of the operating surgeon. In 18 cases, biopsies were available at both timepoints. These biopsies were divided in half, with half placed in formalin for paraffin embedding ahead of histological examination, and the other half placed in RNAlater and snap-frozen in liquid nitrogen for future analysis.

Regular samples of perfusate and ex-situ urine were taken in accordance with the protocol. Sufficient volume was discarded to remove that within the sampling line. Arterial and venous samples taken for blood gas analysis were immediately capped, taken directly to an ABL90-FLEX blood gas analyser, and processed. Perfusate and urine samples were centrifuged, aliquoted, and snap-frozen in liquid nitrogen at the point of collection. These were stored for later analysis at −80 °C.

Post-operative study-specific samples of blood and urine were collected from the recipient daily (post-transplant days 1–4), in accordance with the protocol. Blood was collected into serum separator tubes, centrifuged, and serum stored at −80 °C. 24-h urine collections were obtained. Daily samples were taken from these collections and stored at −80 °C. There were no study-specific samples beyond day 4; all other endpoints were met through routinely measured clinical parameters.

### Clinical outcomes, derived variables, and measures of ex-situ function

**Primary objective—safety and feasibility.** The primary outcome was 30-day graft survival, defined as a functioning graft in a patient who does not require chronic dialysis. Additionally, adverse events were monitored for 12 months from transplant. The severity of events was graded according to the Clavien-Dindo classification. Any organ discards were also logged.

**Secondary outcomes.** Secondary outcomes stated in the protocol were 3- and 12-month graft survival, 30-day, 3- and 12-month patient survival, early measures of graft function (incidence and frequency of the use of dialysis in the first 7 days post-transplant; incidence of functional delayed graft function (fDGF); day 2 creatinine reduction ratio; proteinuria; urine production), incidence of Primary Non-Function (PNF), measures of graft function at 30 days, 3- and 12-

months (eGFR, serum creatinine gradient, and proteinuria), and incidence of acute rejection.

**Exploratory outcomes.** Exploratory outcomes were grouped under three objectives—to study the effects of prolonged ex-situ NMP on ischaemia-reperfusion injury (IRI); to identify biomarkers predictive of post-transplant outcome; and to characterise the performance of the ex-situ preservation system.

The effect of prolonged NMP on IRI was assessed by correlating perfusion duration against a histological assessment of tubular injury, and against post-transplant injury markers. Candidate biomarkers were specified in the protocol in advance. A recently-published systematic review of biomarkers identified in perfusate during hypothermic machine perfusion of the kidney[34], NMP studies published by our own and other leading groups[10,45], studies examining biomarkers for AKI in the context of cardiopulmonary bypass[46], and relevant donor biomarker studies were considered[47]. The final pre-specified biomarker list was NGAL, KIM-1, L-FABP, GST, LDH, AST, IL-18, and cell-free DNA. These biomarkers were correlated against early and late measures of graft function. To characterise the performance of the preservation process, various measures of ex-situ homeostasis, including perfusate biochemistry, haemolysis, and oxygen consumption, were monitored over time during perfusion.

**Derived variables.** Throughout the trial, wherever eGFR was used (donor pre-retrieval kidney function; recipient at 30 day, 3 months, and 12 months post-transplant), it was calculated by the CKD-EPI formula[7]:

$$eGFR = 141 \times \min\left(\frac{SCr}{\kappa}, 1\right)^{\alpha} \times \max\left(\frac{SCr}{\kappa}, 1\right)^{-1.209} \times 0.993^{Age} \quad (2)$$
$$\times 1.018 \,[if\ female] \times 1.159 \,[if\ black]$$

Where SCr is serum creatinine, alpha is −0.329 for females and −0.411 for males, κ is 0.7 for females and 0.9 for males, min is the minimum of Scr/κ or 1, and max is the maximum of Scr/κ or 1.

The UK Kidney Donor Risk Index (2019 version[30]) was used to match trial cases to historical controls. This was calculated as:

$$DRI = \exp\left\{ \begin{array}{l} 0.023 \times (donor\ age - 50) - 0.152 \times \frac{(donor\ height) - 170}{10} \\ + 0.149 \times (history\ of\ hypertension) \\ - 0.184 \times (female\ donor) \\ + 0.190 \times (CMV\ positive\ donor) \\ - 0.023 \times \frac{(offer\ eGFR - 90)}{10} \\ + 0.015 \times (days\ in\ hospital) \end{array} \right\} \quad (3)$$

Functional delayed graft function (fDGF) was defined as either the use of post-transplant dialysis in the first 7 days or a failure of serum creatinine to fall by at least 10% per day for the first 3 days. fDGF was used preferentially as the early outcome for analyses examining ex-situ biomarkers, as it accounts for recipients who were pre-dialysis at the point of transplantation.

Later outcomes, where correlated against ex-situ biomarkers, are reported as deindexed eGFR to reduce the influence of recipient size[48]. Body surface area (BSA) was estimated by the method of Mosteller[49], and deindexed eGFR calculated as:

$$eGFR_{deindexed} = \frac{eGFR_{BSA-indexed} \times BSA_{recipient}}{1.73} \quad (4)$$

The recirculation of urine ex-situ means that a steady-state is reached, and the ratio of the measured creatinine concentration in the

urine and perfusate can be used to calculate creatinine clearance:

$$CrCl_{Ex-situ} = \frac{[Cr]_{urine} \times Q_{urine}}{[Cr]_{perfusate}} \quad (5)$$

Fractional excretion of sodium was calculated as:

$$Fe_{Na} = \frac{[Na]_{urine} \times [Cr]_{perfusate}}{[Na]_{perfusate} \times [Cr]_{urine}} \quad (6)$$

The tubular sodium reabsorption rate was calculated as:

$$Tubular\ reabsorption_{Na} = \left(CrCl \times [Na]_{perfusate}\right) - \left(Q_{urine} \times [Na]_{urine}\right) \quad (7)$$

Oxygen consumption was calculated as previously described in ref. 50:

$$vO_2 = Q_{art} \times \left(\alpha \times (paO_2 - pvO_2) + \beta \times [Hb] \times (S_{art} - S_{ven})\right) + Q_{urine} \times \alpha \times (pvO_2 - puO_2) \quad (8)$$

where Q refers to flow, pa and pv refer to partial pressures in arterial and venous perfusate, pu refers to partial pressure in the urine, S is haemoglobin-oxygen saturation, alpha is the coefficient of solubility of oxygen in plasma, and beta is Huffner's constant (the oxygen carrying capacity of haemoglobin).

## Sample analysis

Biomarkers known to be associated with generic cellular or specific renal injury, or previously reported in the context of ex-situ kidney studies, were specified in the protocol in advance. These were lactate dehydrogenase (LDH), aspartate aminotransferase (AST), neutrophil gelatinase-associated lipocalin (NGAL), kidney injury molecule 1 (KIM-1), liver-type fatty acid binding protein (L-FABP), interleukin 18 (IL-18), glutathione serum transferase (GST), and cell-free deoxyribonucleic acid (cfDNA). Markers of renal function (urine production, creatinine clearance, tubular reabsorption of sodium, oxygen consumption, and on-pump proteinuria) were also measured, as was vascular resistance.

For conventional analytes (including electrolytes, osmolality, creatinine, albumin, insulin, AST, and LDH) samples were analysed using standard techniques, to a validated clinical standard, by the Oxford University Hospitals Biochemistry Department. Other biomarkers in perfusate samples, and NGAL in post-operative serum samples, were analysed using ELISA kits in accordance with the manufacturer's instructions (NGAL, KIM-1, and IL-18: R&D Systems, Inc; GST-Pi and cfDNA: AbCam; L-FABP: Hycult Biotech). Optimisation experiments were conducted for all analytes using pooled samples, at least three dilutions, and calculation of percentage recovery. All samples were analysed in duplicate and quality-controlled; results where there was >30% difference between duplicates were rejected.

Formalin fixed, paraffin embedded biopsies were cut and stained (haematoxylin and eosin, and periodic acid-Schiff) by the Oxford Centre for Histopathological Research. They were read by an independent pathologist blinded to perfusion duration. Features of acute and chronic damage were identified and scored by the reporting pathologist, in biopsies both pre-perfusion and post-reperfusion in the recipient. The following parameters were scored: global glomerulosclerosis (none, <25%, 26–50%, >50%); tubular atrophy/interstital fibrosis (0–5%, 6–25%, 26–50%, >50%); arteriosclerosis (normal, fibroelastosis with the intima thinner than media, fibroelastosis with intimal thickness equal to the media, fibroelastosis with the intima thicker than the media); arteriolar hyalinosis (none, mild, moderate, severe), and acute tubular injury absent, loss of brush border and/or vacuolation of tubular epithelial cells, cell detachment/cellular casts <50%, cell detachment/cellular casts >50% or cortical necrosis. For markers of chronic damage, the highest grade seen per case was used

in further analyses, irrespective of the timepoint of the biopsy. For acute tubular injury, the grade was compared between pre- and post-perfusion biopsies.

For the analysis of perfusate and functional biomarkers, and association with post-transplant outcomes, we chose to evaluate profiles at 2 h and at the end of perfusion. This was because the trial protocol stipulated a minimum perfusion duration (2 h) and a maximum, but did not mandate a specific duration within the permitted interval. Therefore, samples were available for all cases at these timepoints (aside from one case where the 2-h perfusate sample was also the end-perfusion sample). We evaluated profiles at hour 2 rather than earlier in perfusion as we observed that renal blood flow took 2 h to stabilise (Fig. 3A), and particularly for functional measures, we felt that assessment of the kidney in a steady state was important.

## Data management

All study-specific data were collected and managed using REDCap electronic data capture tools hosted at the University of Oxford. REDCap (Research Electronic Data Capture) is a secure, web-based software platform designed to support data capture for research studies. Source data verification was provided by the Surgical and Interventional Trials Unit (SITU).

## Statistical analysis

**Interim analysis.** Reports for the Data Safety and Monitoring Committee (DSMC) were prepared after recruitment of patients 10 and 22, with progression to stages 2 and 3, respectively, contingent on approval from the DSMC and Trial Steering Committee (TSC). In addition, there was a pre-specified stopping rule, with a recruitment pause and ad-hoc review by the TSC and DSMC mandated should two out of any six consecutive recipients experience early graft loss.

**Description of statistical methods.** The final analysis considered all recipients of normothermically-perfused kidneys. 12 patients were recruited but excluded from the analysis (see Fig. 1); all 12 who were withdrawn did not receive kidneys which had been exposed to NMP. One of these 12 withdrawals was related to a protocol deviation—the perfusion protocol was followed correctly, however the observation of profoundly abnormal ex-situ perfusion parameters led to discard of a kidney that otherwise would have been transplanted (see results, and Fig. 4A). There were no protocol deviations precluding inclusion of recipients transplanted with a normothermically perfused kidney in the final analysis. There were three outliers with respect to the length of the second cold ischaemic time (see Results) due to the need for post-perfusion vascular assessment/reconstruction; these cases were included in all analyses.

Descriptive statistics are provided for all primary, secondary, and exploratory outcomes—mean and standard deviations for continuous data where this was normally distributed or median and interquartile range where it was not, and frequency and percentage for categorical data. Data was complete with respect to baseline donor and recipient demographics and clinical follow-up to 1 year for the trial cohort (including patient and graft survival, graft function, and adverse event data).

Where reported, values were compared using chi-squared tests for categorical endpoints, $T$-tests for normally-distributed continuous endpoints, the Wilcoxon signed-rank test for non-normally-distributed continuous endpoints, and Welch's $T$-test where there was evidence of significant heteroscedasticity. All tests were two-sided. Correlation coefficients, where reported, are Pearson's Rho (for normally distributed variables) or Spearman's rank correlation coefficient (for non-normally distributed variables). A $p$ value of 0.05 was used as the threshold for statistical significance, and two-sided tests were used throughout. Unless otherwise specifically indicated in the text, $p$ values were not adjusted for multiple comparisons. Where indicated (see

'Results', specifically concerning the association between the GST-Pi gradient and 12-month eGFR), this was by the method of Bonferroni, accounting for 41 comparisons (biomarkers specified in advance, accounting for where markers were measured at more than one timepoint, and the associated gradient).

Where comparisons were made between the trial cohort and the matched control cohorts, these were univariate—i.e. not adjusted for matching variables. Adjustment for matching factors has been advocated by some authors[51]. Due to our sample size, study design, and number of clinically relevant matching variables, we opted for a simple, unadjusted analysis to inform a conclusion as to whether the technique is sufficiently safe to proceed to phase two trials to test efficacy.

All statistical analysis was performed in R Studio (Version 2023.06.0 + 421).

**Sample size and recruitment.** This was a feasibility trial and not statistically powered. The sample size was based on a compromise between feasibility, precision, and regulatory considerations[52], and an estimated 12-month recruitment timeline.

### Reporting on sex and gender

NKP1 was open to donor organs and transplant recipients of all sexes and genders, and the recruited cohort reflects this (see Table 1). Sex and gender were not explicitly considered by the study design, and so a priori sex- or gender-based analyses have not been performed. Sex was determined as that assigned at birth. Results are not presented disaggregated by sex or gender due to the low numbers inherent in a phase 1 trial, however, some individual-level data are provided in the Source Data file.

### Reporting summary

Further information on research design is available in the Nature Portfolio Reporting Summary linked to this article.

## Data availability

Source data and the study protocol are provided with this manuscript, and all data supporting the findings described in this manuscript are available in the article and in the Supplementary Information. As NKP1 was a single-centre phase 1 trial with a limited number of participants, to protect individual patient confidentiality the source data which is provided is a limited dataset sufficient to reproduce the figures and conclusions presented in this manuscript. Written requests made by qualified medical or scientific professionals, addressed to the corresponding authors, for access to the full trial dataset (individual de-identified participant and perfusion data) will be considered on a case-by-case basis within 3 months of receipt. The purpose of accessing this data must be provided, and access will depend on adequate anonymisation, institutional oversight, review by a qualified panel of experts, and if necessary, an independent ethics committee. A data sharing agreement signed on behalf of both the Sponsor and the receiving organisation would be necessary prior to sharing any data. Source data are provided with this paper.

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

## Acknowledgements

This study/project was funded by the NIHR Invention for Innovation programme (grant number NIHR200022). This award was made to PF (lead applicant) and to C.C., R.P., S.K., A.W. and J.H. as co-applicants. R.D. was directly funded by the award. The views expressed are those of the authors and not necessarily those of the NIHR or the Department of Health and Social Care. The University of Oxford was the Sponsor and direct employer of the lead (R.D.) and chief (P.F.) investigators. The Sponsor's Office did not participate in study design, data collection, analysis, or manuscript writing. We acknowledge the contribution to this study made by the Oxford Centre for Histopathology Research and the Oxford Radcliffe Biobank, which are supported by the University of Oxford, the Oxford CRUK Cancer Centre and the NIHR Oxford Biomedical Research Centre (Molecular Diagnostics Theme/Multimodal Pathology Subtheme), and the NIHR CRN Thames Valley network. We acknowledge the contributions made by the Surgical Intervention Trials Unit and the Oxford Clinical Trials Research Unit. The study was conducted as part of the portfolio of trials in the registered UKCRC Oxford Clinical Trials Research Unit (OCTRU) at the University of Oxford. It has followed their Standard Operating Procedures, ensuring compliance with the principles of Good Clinical Practice and the Declaration of Helsinki and any applicable regulatory requirements. We are very grateful to the patients who participated in this trial. We are also grateful for the contributions of the independent members of the trial oversight committees (Professors Michael Nicholson, Lorna Marson, Colin Wilson, Gabriel Oniscu, Nizam Mamode, Dr Nicholas Torpey, Dr Virginia Chiocchia, and Mr Steve Rogers), and for the support of the Oxford Transplant Centre (Oxford University Hospitals NHS Foundation Trust).

## Author contributions

R.D.: lead investigator, responsible for direct delivery of all aspects of the trial including set-up, recruitment and consent, perfusion, sampling,

sample and data analysis, interpretation, and drafting the manuscript. S.K.: Co-PI, contributed to trial design, oversight, assistance with perfusion, interpretation of results, and drafting the manuscript. J.H., A.W. and R.P.: trial management group members, contributed to trial design, oversight, and drafting the manuscript. J.F.: assisted with clinical perfusions. D.V., J. Barrett, and M.E.: OrganOx engineering and software team responsible for production and maintenance of the perfusion devices. E.C.: trial statistician, responsible for reviewing and verifying the data and analyses. IR: histopathologist responsible for reporting of trial biopsies. T.J., G.A. and J. Brook: clinical biochemistry team responsible for sample analysis. C.C.: trial management group member with overall responsibility for design and construction of the perfusion device, and contributions to trial design, interpretation, and drafting the manuscript. P.F.: chief investigator with overall responsibility for all aspects of the study, including design, delivery, and the manuscript.

## Competing interests

Peter Friend is a founder and is the Chief Medical Officer of OrganOx Ltd Constantin Coussios is a founder and is the Chief Technical Officer of OrganOx Ltd R.D., S.K. and J.F. have received consultancy fees from OrganOx Ltd for work outside the scope of this trial. The remaining authors declare no competing interests.
