## [Transparent Peer Review file · Nature Communications]

Prolonged normothermic perfusion of the kidney – a historically controlled, phase 1 cohort study

Corresponding Author: Dr Richard Dumbill

Version 1:

Reviewer comments:

Reviewer #1

(Remarks to the Author)

This is very well written and interesting manuscript on an important topic and demonstrates a considerable step-change in the longevity of perfusion times for kidney preservation. This is of significant importance to the international field of kidney transplantation demonstrating that extended normothermic preservation is both feasible and safe. There was not any improvement in the graft outcomes but the matched control cohort had generally fitter recipients and poorer quality donors. There were some interesting biomarker results that will inform future utilisation studies.

My main comments regarding the manuscript are:

- 1) There is an impressively low CIT (median 8.5 hours) relative to national average (13 hours)
- 2) The authors need to include some more background information regarding the rationale for including those specific candidate biomarkers.
- 3) More information on the “device failure” reason for the not perfusing & “unsuitable following back benching” was that for complex vascular anatomy or other reasons?
- 4) The phrase “7 transplanted in coventry” should be less centre specific and state “transplanted in our sister centre where the perfusion device was not available”
- 5) With a view to wider adoption of the device can the authors comment on if the PTFE graft will be part of protocol/ device format going forward for arterial cannulation or is there an alternative solution?
- 6) By what mechanics do you think the lactate is clearing on isolated kidney perfusion circuit with recirculating urine? From our groups experience of kidney NMP the lactate usually rises throughout perfusion, it is rare to see any kidneys clear it?
- 7) Why have the authors chosen to perform viability testing at 2 hours when previous case series have reported those figures at 1 hour of NMP?
- 8) Can the authors please elaborate on the reason for the graft loss at 8 months in the NMP group?
- 9) Notably, there are higher than previously reported renal blood flow during NMP (even after dropping the MAP from 90mmHg to 75mmHg) at around 660ml/min on average. This amount of blood flow through a single kidney could be considered supra physiological. Were there leaks/bleeds from biopsy sites to account for this? Do the authors feel those RBF values are safe considering they are higher than what could be expected in situ?
- 10) The acronyms of biomarkers need described in full for their first use. Explanation for the choice of biomarkers should be included as stated earlier
- 11) What are the authors thoughts on the lower rate of rejection in the NMP group?
- 12) Why was antifungal changed during the protocol

Reviewer #2

(Remarks to the Author)

This study aimed to explore the safety and feasibility of NMP with donor kidneys before kidney transplantation, compared to SCS control group. The study was well designed, and the results showed that NMP group had comparable outcomes to a matched control group, although the total preservation time were much longer in NMP group. There are several questions below need to be elucidated. 1. When select the cases in the matched control cohort, why not choose the cases with cold ischemia time similar to total preservation time in NMP group. Longer CIT in control group may show the advantage of outcomes in NMP group. 2. As for the ex-situ biomarker, could you show the results of biomarkers in different timepoint that correlated to outcomes of kidney transplant? 3. In figure 3H, the level of lactate was reduced in perfusion period, however most of NMP cases in our experience, we found lactate level kept rising during perfusion period, could you explain the reason ?

Reviewer #3

(Remarks to the Author)

The authors present the results of a non-randomised study with matched historical controls, examining the outcomes of kidney transplant patients after NMP compared to SCS. They show that the results of both groups are 'comparable', with NMP offering the notable advantage of longer preservation times.

I will focus my review on statistical aspects.

MAJOR COMMENT:

The authors opted for matched historical controls and describe in the protocol and method section how they matched 2 controls for each patient. Their analysis ignore this matching. In light on the (sometimes conflicting) literature on the topic (eg, for example <https://www.bmj.com/content/352/bmj.i969>), could authors discuss their choice?

MINOR COMMENTS:

/1/ MAIN TEXT:

- on line 129, authors mention 'three outliers'. Could they explain what they did of the corresponding observations in further analyses?
- on line 761, authors mentioned missing data was supplemented with data derived from national registry. Could they be more precise? What missing data? %, type of 'imputation'.
- one line 996, authors mention using mean and sd for continuous data but actually often prefer median and IQR.
- on line 1006, typo 'statistical'.
- on line 1006, authors may be willing to mention 'two-sided tests were used everywhere'.
- on line 1008, could the authors mention which analysis consider Bonferroni multiplicity correction? (unclear to me)

/2/ TABLES:

- in Table 1, for skewed continuous outcomes, authors sometimes prefer mean and sd and sometimes median and iqr. any particular reason?

/3/ FIGURES:

In Figures 3 and S5 (as well as different panels of Figure S9-10),

- authors consider different perfusion time ranges (x-axes) when analysing/displaying data. I was expecting 24h to be the upper bound. Any particular reason for selecting different time ranges?
- authors use dashed red lines often without explaining what they correspond to and how they were defined.

In Panel S6A, authors don't seem to specify which test was used. They may be willing to mention it and to prefer a Welch t-test if they used a Mann-Whitney-Wilcoxon in light of the strong heteroscedasticity.

In Panels S6B+D (as well as different panels of Figure S9-10), same comment as above regarding heteroscedasticity. In Panel S6D, visually, the p-value seems small. The authors may be willing to make a sanity check and re-run this analysis?

In Panels S6E-S6J (as well as different panels of Figure S9-10): authors seemingly show the linear regression line - which has a 1:1 relationship with the Person's correlation estimate in simple cases, like here - but (inconsistently) report the Spearman correlation (and inference) in some cases. Any particular reason? [authors mention on line 1004-5 that spearman

would be used for non-normally distributed data, which may leave some readers wonder why they fit a linear regression]

Version 2:

Reviewer comments:

Reviewer #1

(Remarks to the Author)

A thorough response to all reviewers comments

Reviewer #3

(Remarks to the Author)

I apologise for the delay in returning my review.

Overall, I am satisfied with the authors' responses to the reviewer's points.

I have a minor request for the authors:

I would like them to mention at least one reference discussing the main approaches used to analyse case-control studies, as this topic is anything but trivial. I found the following reference by Neil Pearce (who I am not related to) useful:

<https://doi.org/10.1136/bmj.i969>. I would also like them to summarise the rationale for their choice of a simple analysis. This could align with their response: "We are therefore of the opinion that a simple analysis that informs a conclusion as to whether the technique is sufficiently safe to proceed to phase two trials to test efficacy is preferable."

Whilst I do not dispute the conclusion, I would like readers to understand that this is possibly not the perfect analysis for the reasons discussed by Prof Pearce.

Best, DLC

made.

Prolonged normothermic perfusion of the kidney – a historically controlled, phase 1 cohort study – response to reviewers

REVIEWER COMMENTS

Reviewer #1 (Remarks to the Author):

This is very well written and interesting manuscript on an important topic and demonstrates a considerable step-change in the longevity of perfusion times for kidney preservation. This is of significant importance to the international field of kidney transplantation demonstrating that extended normothermic preservation is both feasible and safe. There was not any improvement in the graft outcomes but the matched control cohort had generally fitter recipients and poorer quality donors. There were some interesting biomarker results that will inform future utilisation studies. My main comments regarding the manuscript are:

1) There is an impressively low CIT (median 8.5 hours) relative to national average (13 hours)

The low CIT seen in our trial reflects the fact that it is more straightforward to start normothermic machine perfusion than it is to transplant the kidney. In our institution (and many others) it is common practice to defer transplant until the morning, if the kidney arrives after midnight. We did not defer the start of perfusion in any of the cases in our trial – we began perfusion as quickly as possible on arrival of the kidney at our centre. This explains the low CIT.

2) The authors need to include some more background information regarding the rationale for including those specific candidate biomarkers.

We have added additional detail and references to the section Methods/ Clinical outcomes, derived variables, and measures of ex-situ function/ Exploratory outcomes to explain the rationale for selecting the specified biomarkers.

3) More information on the “device failure” reason for the not perfusing & “unsuitable following back benching” was that for complex vascular anatomy or other reasons?

The device failure was a software error that meant that the pump could not be started. Of the three kidneys deemed unsuitable following back benching, one had not been flushed adequately at retrieval and the other two had significant arterial dissections. **These details have been added to the Figure 1 legend.**

4) The phrase “7 transplanted in coventry” should be less centre specific and state “transplanted in our sister centre where the perfusion device was not available”

We have updated Figure 1 accordingly

5) With a view to wider adoption of the device can the authors comment on if the PTFE graft will be part of protocol/ device format going forward for arterial cannulation or is there an alternative solution?

We felt PTFE graft to be a safe and secure method of cannulation, that allowed inclusion of all kidneys irrespective of vascular anatomy. It represents a trade-off (higher technical complexity for greater flexibility, security, and minimal alteration to the donated kidney) that we felt was appropriate for a phase 1 trial. Beyond our trial other methods of cannulation (for example patch clamps and direct cannulation) are feasible for selected kidneys – and in particular patch clamps are likely to be suitable for most kidneys with single arteries and good-quality aortic patches. We envisage the final solution to be a tool kit that contains these various options, for the perfusionist to select from as required. **We have added this detail to the Discussion.**

6) By what mechanics do you think the lactate is clearing on isolated kidney perfusion circuit with recirculating urine? From our groups experience of kidney NMP the lactate usually rises throughout perfusion, it is rare to see any kidneys clear it?

This is an interesting observation, and we have noted this difference between our results and those obtained by others with respect to lactate clearance. There are several possible explanations:

- 1) Cortical hypoperfusion – a possible intra-renal Cori cycle has previously been proposed (Bankir, 2021) with lactate produced by glycolysis in the relatively hypoxic medulla used by proximal tubular epithelial cells in the cortex to produce the glucose used deeper in the medulla. Relative hypoperfusion of the cortex could therefore be expected to lead to rising lactate levels, as insufficient gluconeogenesis occurs to deal with lactate generated by the medulla. As noted in point (9) below the renal blood flows seen in our trial were substantially higher than reported previously, and so cortical hypoperfusion in other reports is a plausible mechanism by which lactate might rise.
- 2) Perfusate composition – the balance of supplied metabolic substrate and insulin concentration might be important in determining net lactate production. We did not supply amino acids, which differs from other described protocols. Figures 3 H and I indicate the glucose and insulin concentrations observed with our protocol. Further work needs to be done to elucidate whether this difference in lactate behaviour is due to differences in metabolic substrate or regulation of metabolism.
- 3) Renal injury, graft quality, and cold ischaemia – Selzner's group have reported that in a porcine model subject to varying warm ischaemia lactate falls during 8 hours of NMP, however the rate of decline is slower with longer warm ischaemic times (Kaths 2018). The donors used in the study were young, healthy animals and there was minimal cold ischaemic time. They did not use urine recirculation. Even with an hour of warm ischaemia, lactate still fell. This does not explain the rise in lactate seen by other groups, however it may be the case that with human organs allocated to research because of concerns that they are untransplantable, the combination of the marginal nature of the kidney and often prolonged cold ischaemic times leads to a rise rather than a fall in perfusate

lactate. Again, further work needs to be done to elucidate the contributions of these factors to ex-situ lactate dynamics.

We did not see any correlation between lactate clearance and clinical outcome. **We have added these points to the manuscript (Discussion).** We are in agreement with the reviewer that this is an area of future interest, and it should be the subject of further investigation.

7) Why have the authors chosen to perform viability testing at 2 hours when previous case series have reported those figures at 1 hour of NMP?

Previous case series have been more limited with respect to perfusion duration than in our trial. Whilst the Hosgood-Nicholson Quality Assessment Score has been reported at one hour, and the Cambridge paradigm is to date the most widely studied, we note that in the most recent randomised controlled trial of this implementation of renal NMP (Hosgood et al. 2023) the QAS was not associated with clinical outcome. Other authors have reported biomarkers at other timepoints (e.g. Selzner's group report lactate at one and two hours in both their porcine pre-clinical studies, and their single clinical paper; Rijkse et al. report lactate and renal blood flow at 2 hours in their pilot study of 2 hours of NMP prior to transplantation in the Eurotransplant Senior program).

We noted that in our trial renal haemodynamics (Figure 3A) were not stable until 2 hours. We suspect that biomarker interpretation is to an extent dependent on the perfusion device and perfusate used, and so comparison to other reported case series is challenging regardless of the timepoint selected. As two hours was the minimum perfusion duration during our trial, we had biomarker data at two hours for all patients in the trial – and so used this timepoint. We agree with the reviewer that the best timepoint to perform viability assessment remains to be determined. **We have added to our Results section to expand on our rationale for selecting this timepoint. We have also added to the discussion to highlight that further work on viability assessment needs to be done.**

8) Can the authors please elaborate on the reason for the graft loss at 8 months in the NMP group?

The consensus opinion of the clinical and trial teams was that this was due to progression of donor diabetic disease.

The donor was a 30-year-old DBD donor, who had died in diabetic ketoacidosis. Initially the recipient had excellent primary function (creatinine 892 -> 636 -> 403 -> 314, no post-transplant dialysis), however creatinine was stuck around 250 from day 4 onwards and no real diagnosis reached; a biopsy at this point showed donor diabetic changes only, but minimal chronic damage and no rejection. Similar donor diabetic lesions were present on a pre-implant biopsy however donor renal function was normal (creatinine 55, single + for protein on a dipstick, no donor urine protein:creatinine ratio available). We did note that heavy proteinuria was present throughout the post-transplant period, and the recipient's primary renal disease was unknown (it is possible that this was FSGS, but this was not proven). At 8 months post-transplant the patient presented with progressive renal failure and fluid overload. There was a poor response to high-dose

furosemide, and a magnetic resonance angiogram ruled out transplant renal artery stenosis. Again, there was no evidence of rejection on a repeat biopsy – only diabetic changes. Unfortunately, the recipient returned to maintenance dialysis shortly after.

We have added detail on this to the manuscript (Results, Primary and Secondary Outcomes).

9) Notably, there are higher than previously reported renal blood flow during NMP (even after dropping the MAP from 90mmHg to 75mmHg) at around 660ml/min on average. This amount of blood flow through a single kidney could be considered supra physiological. Were there leaks/bleeds from biopsy sites to account for this? Do the authors feel those RBF values are safe considering they are higher than what could be expected in situ?

We agree that renal blood flow being higher than previously reported during NMP is interesting, and potentially a concern. We don't feel that there was a significant rate of bleeding – a single needle core biopsy was taken pre-perfusion, and this was sutured before NMP was started. Given the lack of ex-situ neuro-hormonal control (for example no sympathetic innervation, circulating catecholamines, or intact renin-angiotensin system) we don't believe that low ex-situ vascular resistance is surprising.

A commonly-used figure (for a 70kg adult with a cardiac output of 5L/min) is that total renal blood flow is 1L/min, or 500mL/min per kidney (although this varies depending on a multitude of factors include age and sex – review available here: <https://doi.org/10.1007/s10334-023-01126-7>). The mean±standard deviation for renal blood flow measured at two hours was 744±189mL/min in the 90mmHg group, and 511±162mL/min in the 75mmHg group – which is similar to previously-reported in-vivo values.

We agree that in future consideration should be given as to whether flow-capping is necessary (i.e. down-titration of arterial pressure if physiological renal blood flow is exceeded).

We have added comment on the observation that our observed renal blood flows were higher than those reported previously to the Discussion.

10) The acronyms of biomarkers need described in full for their first use. Explanation for the choice of biomarkers should be included as stated earlier

Expansions for biomarker acronyms have been added to the body of the manuscript. Explanation for the choice of biomarkers has been added to the Methods section, as explained above.

11) What are the authors thoughts on the lower rate of rejection in the NMP group? This is an interesting observation, however the numbers in our study are too small to provide much comment. It is particularly interesting in the context of the COMPARE trial results (Jochmans et al. 2020), which suggested that oxygenation during hypothermic machine perfusion might lead to less rejection. It is plausible NMP reconditions the endothelium to some extent, and that less endothelial injury results in a less

immunogenic graft, however further studies are needed to address this. **We have not made any changes to the manuscript in response to this point.**

12) Why was antifungal changed during the protocol

This was due to availability – midway through the trial the supply of micafungin became restricted, and the price increased to over £500 per dose; we therefore switched to an alternative echinocandin. **We have not made any changes to the manuscript in response to this point.**

Reviewer #2 (Remarks to the Author):

This study aimed to explore the safety and feasibility of NMP with donor kidneys before kidney transplantation, compared to SCS control group. The study was well designed, and the results showed that NMP group had comparable outcomes to a matched control group, although the total preservation time were much longer in NMP group. There are several questions below need to be elucidated. 1. When select the cases in the matched control cohort, why not choose the cases with cold ischemia time similar to total preservation time in NMP group. Longer CIT in control group may show the advantage of outcomes in NMP group. 2. As for the ex-situ biomarker, could you show the results of biomarkers in different timepoint that correlated to outcomes of kidney transplant? 3. In figure 3H, the level of lactate was reduced in perfusion period, however most of NMP cases in our experience, we found lactate level kept rising during perfusion period, could you explain the reason ?

Many thanks for these comments. In response to each in turn:

- 1) We chose to match controls on the basis of cold ischaemia time rather than total preservation time as the purpose of this trial (as a phase 1, first-in-human evaluation of this particular device and protocol) was to evaluate the safety and feasibility of prolonged renal NMP, rather than to test its efficacy. We felt that for the primary analysis it was important to set a high bar when making comparisons to a non-randomised control cohort, to avoid drawing false-positive conclusions at this early stage. We did conduct an exploratory analysis comparing to a secondary matched cohort where matching was done on the basis of total preservation time, as suggested – this is reported in the supplementary material (SDC figure S4, and SDC tables S7 and S8). We did not see any differences in this analysis. **We have added to the discussion in the manuscript to underline the reason why we chose to match on the basis of cold ischaemia time.**
- 2) Timepoint of the biomarker analysis – the design of the trial was such that a minimum perfusion duration (2 hours) was specified, rather than a target. Therefore, the only sample collection time-points where we have data for all 36 cases are hours 1, 2, and the end of perfusion (variable). We chose to use hour 2 for these analyses as the renal blood flow data (figure 3A) shows that steady-state perfusion was not reached until this point; for the assessment of ex-situ function, we felt that it was important that the kidney was in a steady state. We have provided some other biomarker data from other timepoints - supplementary data SDC figure provides additional biomarker analyses measured at the end of perfusion. **We have added an explanation of this to the Methods.**
- 3) The observations regarding lactate falling during perfusion are indeed interesting and were also raised by another reviewer – see response to point (9), reviewer 1. **We have added these points to the manuscript (Discussion).**

Reviewer #3 (Remarks to the Author):

The authors present the results of a non-randomised study with matched historical controls, examining the outcomes of kidney transplant patients after NMP compared to SCS. They show that the results of both groups are 'comparable', with NMP offering the notable advantage of longer preservation times.

I will focus my review on statistical aspects.

MAJOR COMMENT:

The authors opted for matched historical controls and describe in the protocol and method section how they matched 2 controls for each patient. Their analysis ignore this matching. In light on the (sometimes conflicting) literature on the topic (eg, for example <https://www.bmj.com/content/352/bmj.i969>), could authors discuss their choice?

Thank you for this observation. We matched controls to cases on the basis of four variables (donor risk index, cold ischaemia time, donor type, and induction immunosuppression) and the trial cohort consists of 36 patients – we didn't feel that we have adequate numbers to report analyses adjusted for four matching factors. Our matching strategy has led to very close matching between the cases and controls with respect to the matching variables (table 1) – indeed two of the variables were categorical, and the matches were absolute. The most important consideration though is the purpose of the study – our objective was to evaluate the safety and feasibility of prolonged NMP, not to test its efficacy. Our reason for including a control cohort in this early-phase study was simply that gross differences in post-transplant graft function (versus similar transplants exposed to standard care) would represent a safety concern, and so it was important to gather data on graft function in a relevant control cohort. However, we did not seek to test the efficacy of NMP as a preservation modality versus static cold storage – this should be considered by later-phase studies, that are preferably randomised and adequately powered. We are therefore of the opinion that a simple analysis that informs a conclusion as to whether the technique is sufficiently safe to proceed to phase two trials to test efficacy is preferable.

We have therefore removed the 'Comparison' column from Table 2 and have provided simple measures of location and spread for both groups only, without formal statistical comparison. We feel that this approach simplifies display of the data and reduces the risk that our study is read as a formal comparison of the two techniques. This doesn't change our conclusions.

MINOR COMMENTS:

/1/ MAIN TEXT:

- on line 129, authors mention 'three outliers'. Could they explain what they did of the corresponding observations in further analyses?

We included these cases in all further analyses. The second ischaemic period (i.e. the cold time between the end of NMP, and reperfusion in the recipient) will undoubtedly vary should this technology progress to further evaluation, and eventual clinic use. There will always be cases where post-perfusion, pre-implant vascular reconstruction is necessary. It is therefore that these cases are included with the trial cohort. **We have added this explicitly to the Methods section (Description of statistical methods).**

- on line 761, authors mentioned missing data was supplemented with data derived from national registry. Could they be more precise? What missing data? %, type of 'imputation'.

We have clarified this statement in the revised manuscript. What was meant was that where data was missing in our local database, it is usually because patients have moved away from our centre back to local referring renal units. Patient and graft outcomes should be reported for all patients to the national transplant registry; however this doesn't always occur. We merged our local database with the national registry to obtain as complete a dataset as possible concerning graft and patient outcomes for the control cohorts. Where outcomes were missing in both, data was treated as truly missing. No imputation was performed. **Details of any missing data have been added to the legend for table 2.**

- on line 996, authors mention using mean and sd for continuous data but actually often prefer median and IQR.

We have updated this statement in the revised manuscript to explain that where data were normally distributed we have reported means and standard deviations; where data were not normally distributed we have reported medians and interquartile ranges.

- on line 1006, typo 'statistical'.

Thank you, we have corrected this in the revised manuscript

- on line 1006, authors may be willing to mention 'two-sided tests were used everywhere'.

Thank you, this is true, and we have added this statement.

- on line 1008, could the authors mention which analysis consider Bonferroni multiplicity correction? (unclear to me)

Thank you, we have included the details in the revised manuscript.

/2/ TABLES:

- in Table 1, for skewed continuous outcomes, authors sometimes prefer mean and sd and sometimes median and iqr. any particular reason?

The normality of all variables was tested. Where the distribution was not significantly different from normal the mean and standard deviation was reported; where it was significantly different from normal the median and IQR was reported. **As suggested above we have updated the methods section in the revised manuscript to explain this.**

/3/ FIGURES:

In Figures 3 and S5 (as well as different panels of Figure S9-10),

- authors consider different perfusion time ranges (x-axes) when analysing/displaying data. I was expecting 24h to be the upper bound. Any particular reason for selecting different time ranges?

The reason for selecting different time ranges is small numbers. The subplots referenced in Figure 3 show haemodynamic and biochemical parameters measured over time. The number of kidneys perfused for over 12 hours was small (n=4), and so data have not been shown for these timepoints. **We have added an explanation of this to the figure legend.** We agree that the relevant panels of Figure S10 were inconsistent (only showed data up to 6 hours rather than 12) and we **have corrected this in the revised Supplementary Digital Content document.**

- authors use dashed red lines often without explaining what they correspond to and how they were defined.

Thank you – these dashed red lines indicate normal physiological ranges, and **we have added an explanation and numbers used to each relevant figure legend.**

In Panel S6A, authors don't seem to specify which test was used. They may be willing to mention it and to prefer a Welch t-test if they used a Mann-Whitney-Wilcoxon in light of the strong heteroscedasticity.

Thank you – **we have added these details.** You had correctly identified that we had used a Mann-Whitney-Wilcoxon test, and **we have changed this to a Welch t-test** in view of your observation regarding heteroscedasticity. The conclusions have not changed.

In Panels S6B+D (as well as different panels of Figure S9-10), same comment as above regarding heteroscedasticity. In Panel S6D, visually, the p-value seems small. The authors may be willing to make a sanity check and re-run this analysis?

Thank you – again, in the revised document **we have changed from Wilcoxon tests to Welch's T-tests where relevant.** The p-value for panel S6D you reference has increased slightly as a result.

In Panels S6E-S6J (as well as different panels of Figure S9-10): authors seemingly show the linear regression line - which has a 1:1 relationship with the Person's correlation estimate in simple cases, like here - but (inconsistently) report the Spearman correlation (and inference) in some cases. Any particular reason? [authors mention on line 1004-5 that spearman would be used for non-normally distributed data, which may leave some readers wonder why they fit a linear regression]

Thank you for this observation. Some variables – particularly perfusion/ preservation times – are highly skewed and considering the data raw lends undue weight to outliers. We agree that presentation of the Spearman correlation is inconsistent alongside a linear regression, **and so have changed these plots where relevant to display ranked data.**

** See Nature Portfolio's author and referees' website at www.nature.com/authors for information about policies, services and author benefits.

This email has been sent through the Springer Nature Tracking System NY-610A-NPG&MTS

Confidentiality Statement:

This e-mail is confidential and subject to copyright. Any unauthorised use or disclosure of its contents is prohibited. If you have received this email in error please notify our Manuscript Tracking System Helpdesk team at <http://platformsupport.nature.com> . Details of the confidentiality and pre-publicity policy may be found here <http://www.nature.com/authors/policies/confidentiality.html>
Privacy Policy | Update Profile

Prolonged normothermic perfusion of the kidney – a historically controlled, phase 1 cohort study – response to reviewers (2)

REVIEWERS' COMMENTS

Reviewer #1 (Remarks to the Author):

A thorough response to all reviewers comments

Thank you very much indeed. We are very grateful for your attention to our work.

Reviewer #3 (Remarks to the Author):

I apologise for the delay in returning my review. Overall, I am satisfied with the authors' responses to the reviewer's points. I have a minor request for the authors:

I would like them to mention at least one reference discussing the main approaches used to analyse case-control studies, as this topic is anything but trivial. I found the following reference by Neil Pearce (who I am not related to) useful:

<https://doi.org/10.1136/bmj.i969>. I would also like them to summarise the rationale for their choice of a simple analysis. This could align with their response: "We are therefore of the opinion that a simple analysis that informs a conclusion as to whether the technique is sufficiently safe to proceed to phase two trials to test efficacy is preferable." Whilst I do not dispute the conclusion, I would like readers to understand that this is possibly not the perfect analysis for the reasons discussed by Prof Pearce.
Best, DLC

Thank you very much indeed for your further comments. We agree that selecting an approach to the analysis of a case-control study such as this is not trivial. We have included this reference in our revised manuscript as requested. We have also summarised our rationale for our choice of a simple analysis as suggested/ requested. These amendments can be found in the section Methods/ Statistical analysis/ Description of statistical methods.